# Improving Interpersonal Communication by Simulating Audiences with Language Models

## Abstract

How do we communicate with others to achieve our goals? We use our prior experience or advice from others, or construct a candidate utterance by predicting how it will be received. However, our experiences are limited and biased, and reasoning about potential outcomes can be difficult and cognitively challenging. In this paper, we explore how we can leverage Large Language Model (LLM) simulations to help us communicate better. We propose the Explore-Generate-Simulate (EGS) framework, which takes as input any scenario where an individual is communicating to an audience with a goal they want to achieve. EGS (1) *explores* the solution space by producing a diverse set of advice relevant to the scenario, (2) *generates* communication candidates conditioned on subsets of the advice, and (3) *simulates* the reactions from various audiences to determine both the best candidate and advice to use. We evaluate the framework on eight scenarios spanning the ten fundamental processes of interpersonal communication. For each scenario, we collect a dataset of human evaluations across candidates and baselines, and showcase that our framework's chosen candidate is significantly preferred over popular generation mechanisms including Chain-of-Thought. Using a multi-level model, we find that audience simulations achieve significant agreement with human raters. Finally, we demonstrate the generality of our framework by applying it to real-world scenarios described by users on web forums. Through evaluations and demonstrations, we show that EGS enhances the effectiveness and outcomes of goal-oriented communication across a variety of situations, thus opening up new possibilities for the application of large language models in revolutionizing communication and decision-making processes.

## 1 Introduction

We communicate with others in order to achieve our goals: to make friends, to accomplish tasks, or simply to convey our intentions (Grice, 1975; Sperber & Wilson, 1986). However, it can be hard to find the right words to achieve those goals. Consider a scenario where you are trying to get a discount on an item by haggling with its vendor. There are many strategies that you could use to gain an edge, including complimenting the item, offering to buy multiple items for a discount, or even describing your financial situation and asking them to take pity. With so many potential options, it's difficult to correctly decide which strategy to choose. This problem is not confined to bargaining—everyday communication requires us make choices about what approaches to adopt, whether we are making friends, impressing others, or navigating romantic conflicts.

Given a communication scenario, how do we decide which strategies to employ? Often, we rely on heuristics such as our prior experience (Schacter et al., 2007) or on advice we receive from others (Yaniv, 2004). When we have more time to make careful decisions, we may even play out possible candidates in our minds, simulating the reaction of an imaginary listener and using their imagined reaction to guide our choice (Atance & O'Neill, 2001). This idea is formalized in the Rational Speech Act (RSA) model (Goodman & Frank, 2016), which explains people's communication choices in terms of speakers simulating listeners as rational interpreters of possible candidate utterances. However, both our experiences and the advice of others are biased by the information we are exposed to, making our heuristics and simulations imperfect and resulting in suboptimal communication outcomes (Gilbert & Wilson, 2007). Moreover, reasoning about others' potential reactions can be time-consuming and cognitively challenging (Gilbert et al., 1988).

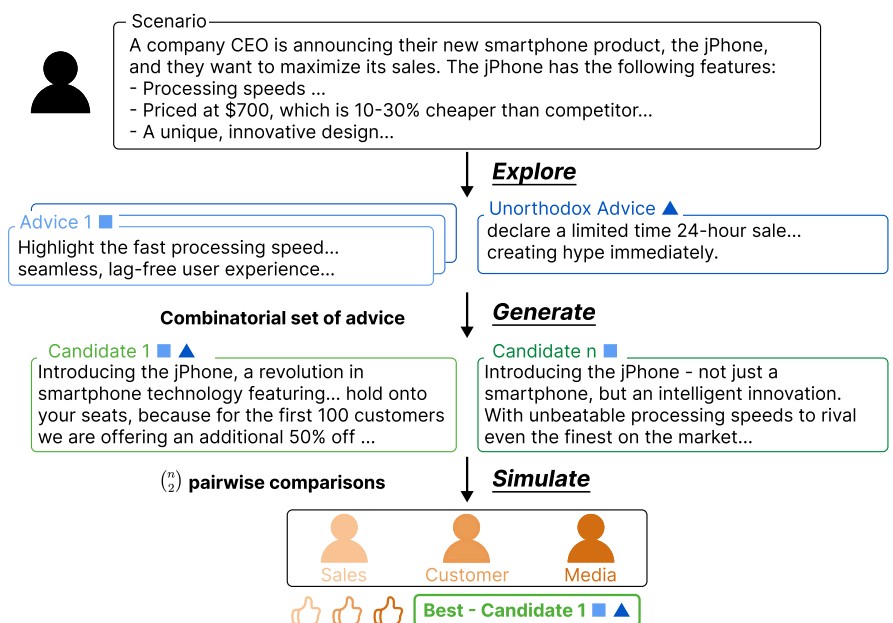

Figure 1: Given a scenario and goal, EGS generates the best candidate message by simulating stakeholders using an LLM. It *explores* pieces of advice that might help, *generates* candidates conditioned on subsets of advice, and *simulates* audience members who evaluate the various candidates.

Inspired by the newfound capacity of LLMs to simulate agents (Park et al., 2023), we propose the Explore-Generate-Simulate (EGS) framework, which supports people in exploring communication strategies and developing message candidates while offloading the cognitively challenging simulation of audience reactions. More precisely, given an arbitrary communication scenario, EGS first *explores* the space of possible responses by using an LLM to produce both normal and creatively unorthodox advice relevant to the scenario. Next, it *generates* communication candidates by conditioning an LLM on various subsets of the advice. Finally, it *simulates* the reception of each candidate by having the LLM take on the perspectives of key audiences. Using these simulations, we can determine which candidates and advice are best suited for achieving the communicator's goal. By construction, our framework also lets us study whether LLMs, conditioned on the scenario and audience description, can effectively simulate audience reactions.

To evaluate this framework, we construct eight diverse scenarios that span the ten fundamental processes of interpersonal communication (Berger & Roloff, 2019) and a variety of communication modalities, relationships, and settings (see Appendix A.1 and A.2). The scenarios include an airline representative speaking to the press after a plane crash, a college student trying to write their profile on a dating app, and even everyday situations like a barista interacting with a customer. We collect human judgments on the effectiveness of each candidate message and compare how EGS performs against non-simulation baseline methods, including GPT-4 zero-shot and Chain-of-Thought (CoT). Finally, we analyze the agreement between humans and simulated audiences using real world scenarios drawn from the Stanford Human Preferences (SHP) dataset (Ethayarajh et al., 2022).

In both evaluations, our approach convincingly outperforms their respective baselines. We also find significant agreement between simulated preferences and human scores across three analyses. We provide qualitative examples, showing LLM-generated messages at each step and highlighting our design choices. Viewing LLMs as a library of shared experiences, our simulation approach draws on this library to integrate individual experiences to ultimately help people communicate better.

## 2 RELATED WORK

For an extended related work discussion on prompt engineering and human preferences, we refer the reader to Appendix D.

**Interpersonal relationships.** Research in social psychology views interpersonal communication from the perspective of ten fundamental processes that underlie social interaction (Berger & Roloff, 2019). These processes may be present regardless of the social context within which communication occurs (see Appendix A.1 for a list). Grice identified a set of maxims surrounding the problem of how a cooperative speaker should choose what to say, such as truthfulness or relevance (Grice, 1957; 1975; 1989). Classical formal models of communication view cooperative communication as information transfer between speaker and listener (de Saussure, 1916; Lewis, 1969; Shannon, 1948), with a focus on aligning the true state between agents as the goal of communication (Stalnaker, 1978). The Rational Speech Act (RSA) framework (Frank & Goodman, 2012; Goodman & Frank, 2016) draws on both, modeling informative speakers as aiming to reduce the listener's uncertainty over the true world state, assuming listeners make rational inferences from the utterances they hear.

**Simulations in the human mind.** The act of projecting oneself into the future to pre-experience an event is formalized in cognitive science as "episodic future thinking" (Atance & O'Neill, 2001). At the neuropsychological level, brain regions traditionally associated with memory are similarly engaged when people imagine future experiences (Schacter et al., 2007). Szpunar (2010) covers the close relation between episodic future thought and the ability to remember personal episodes from one's past. In cognitive psychology, Klein et al. (2010) find evidence that a goal of long-term memory is to store information about the past to plan for the future. Thus, if we consider LLMs as encoding an aggregation of personal experiences across a large subset of human society, they may also have the capacity to simulate epsiodic future thought. Furthermore, as LLMs are theoretically able to take experiences and infer and represent properties of an agent likely to have had those experiences (Andreas, 2022), they may be able to simulate episodic future thought from the perspective of an average agent with a particular set of properties, leading to the possibility of realistic LLM-simulated audiences.

**Agent simulations.** LLM simulations are seen as an opportunity to expand research in computational social science (Ziems et al., 2023). Although rule-based simulations have traditionally been used to study social phenomena, they are limited in expressivity (Schelling, 1971; Easley & Kleinberg, 2010). LLMs can potentially simulate more complex interactions that are harder to codify. They can also be explicitly conditioned to simulate individuals with goals and objectives (Jones & Steinhardt, 2022; Koralus & Wang-Maścianica, 2023; Liu & Shah, 2023). This was recently used to simulate social computing systems (Park et al., 2022), rollout their members' interactions (Park et al., 2023), generate public opinion (Chu et al., 2023), and even produce individualized subjective experience descriptions (Argyle et al., 2023). However, there are uncertainties in using LLM simulations due to their unpredictability (Salganik et al., 2006). LLMs exhibit higher homogeneity of opinions than humans (Argyle et al., 2023; Santurkar et al., 2023), and combining LLMs with human samples is essential to avoid algorithmic monoculture, leading to collapsed generations that constitute only a limited set of perspectives (Kleinberg & Raghavan, 2021; Bommasani et al., 2021).

## 3 THE EXPLORE-GENERATE-SIMULATE (EGS) FRAMEWORK

The EGS framework takes as input an arbitrary scenario where an individual is communicating to an audience with a goal they want to achieve. As output, it suggests a candidate message and set of advice to help the individual achieve their goal.

**Example.** An example input may be:

*A company CEO is announcing their new smartphone product, the jPhone, and they want to maximize its sales. The jPhone has the following features . . . Right now, they are about to give a quick 30-second presentation about the jPhone, broadcasted to major television channels.*

Here, the individual is the CEO, and their goal is to *maximize the sales of the jPhone*. Given the input, we perform three steps:

In the **Explore** step, EGS asks an LLM to generate a diverse set of advice for the scenario. For example, it might suggest:

> *Highlight the fast processing speed and seamless, lag-free user experience.*

In the **Generate** step, for each set of advice, EGS asks an LLM to generate candidates for the communication. For the advice above, it might generate:

*Introducing the jPhone. With unbeatable processing speeds to rival even the finest on the market . . .*

In the **Simulate** step, EGS first asks an LLM to generate a list of stakeholder audience profiles, each with a unique description and perspective. For the above scenario, a stakeholder might be *media outlets*, and their perspective:

> *Your job is to listen to the CEO's presentation, understand the key features and selling points of the smartphone, and relay the information to the public. . .*

EGS uses simulated audience perspectives to evaluate each candidate, with the metric as the likelihood and magnitude to which the candidate achieves the communicator's goal.

**Why is default LLM prompting insufficient?** When using current LLM methods such as CoT, generated candidates often lack diversity in their wording and approach. Furthermore, the communicator might lack the perspectives of the important audiences in order to accurately evaluate one message candidate against another. Additionally, as we broaden the search space with many potential candidates, the user can get easily overwhelmed when deciding between the options, requiring us to build a more scalable method of making judgements between potential message candidates.

Our EGS framework addresses each of these challenges, and it is modular such that each step can be implemented flexibly based on one's specific scenario. We now summarize each EGS component.

## 3.1 EXPLORE

The purpose of the *Explore* step is to expand the space of possible candiate generations. This step generates a list of distinct pieces of advice to later condition the candidate generation upon. We follow existing literature in Social Psychology, which finds that people recall useful advice (Yaniv, 2004) or prior experiences (Schacter et al., 2007) when considering their next action. Similarly, *Explore* generates relevant pieces of advice that will be useful for the next stage of the framework.

Additionally, EGS prompts the LLM to generate "unorthodox but potentially helpful" advice to increase the diversity of candidates. In the example above, GPT-4 generates the unorthodox advice:

> *In your 30-second presentation, declare a limited time 24-hour sale where the first 100 customers get the phone at an additional 50% off, creating hype and urgency to buy immediately.*

We find that unorthodox advice generated by GPT-4 are clever and creative (see Section 4), while also improving the downstream candidates in 4 of 8 scenarios (see Appendix B.8). We theorize that this is because LLMs are exposed to a large trove of human experiences, allowing it to catch successful strategies that are only employed by smaller communities or even just a few individuals.

## 3.2 GENERATE

The *Generate* step seeks to create reasonable candidates for the communication guided by advice from the Explore step. EGS forms combinatorial subsets of the advice generated from the previous step, and each subset is used to generate a few candidate messages.

Following Park et al. (2023), we utilize the "inner voice" of the communicator to condition generated candidates on their assigned advice set, as it makes the LLM more likely to treat the statement as a directive. Using this, we frame the advice set as what the individual remembers in the moment:

> *You remember a few pieces of advice: . . . You decide to focus on using these pieces of advice during your presentation.*

Conditioning on a combinatorial spread of advice further expands the explored solution space. This allows each candidate to incorporate orthogonal advice concepts, which we find leads to better performance through an ablation (see Appendix B.10). We also conduct a preliminary investigation into generating candidates conditioned on specific audiences in addition to advice, and find that they are not preferred over those generated with only advice (see Appendix B.12).

A key feature that enables combinatorial advice is the scalability of EGS. Using LLMs, we can generate and compare between candidates at much larger scale compared to human thought, allowing for more sophisticated explorations around our choices. We showcase the quality of the generated candidates through both a demonstration (Section 4) and quantitative analyses (Section 5).

### 3.3 SIMULATE

The Simulate step consists of two parts. First, EGS generates a list of key audiences who have influence over the communicator's goal, and constructs profile descriptions for each. Next, EGS simulates the reactions of these audiences to each candidate message, and aggregates the results to determine which candidate is best for achieving the goal of the communicator.

For each audience, EGS asks the LLM to construct 1) a description of the scenario and reception of a candidate message from their point of view, 2) the appropriate question to ask for how their reaction to each candidate would directly impact the communicator's goal, and 3) a weight for the audience's relative importance. For the *media outlets* audience in the example, the question generated was:

> *In which scenario would you be more likely to give more media coverage and promotion towards the jPhone?*

We aggregate audience evaluations and get the best candidate and advice using a simple weighted sum. For details on the audience generation and examples, please see Appendix A.5. Motivated by literature on cognitive choices (Gates et al., 2020), we provide other audience aggregation options and a brief analysis in Appendix B.13. We consider aggregation via generating a new candidate using the top-performing candidates and their comparisons in Appendix B.14.

Since candidates can be very close in quality, LLMs can lack granularity when giving ratings to individual candidates (Qin et al., 2023). Thus, we ask simulated audiences to use pairwise comparisons instead. We provide simulated audiences with two scenarios, one representing each candidate in the pairwise comparison (see Appendix A.6), and ask it to reason about which is better before providing an outcome $o \in \{$"prefer scenario 1", "prefer scenario 2", "tie"$\}$. Once we have outcomes for each audience, we aggregate scores across audiences to get the best candidate $c^*$. We perform an ablation to show that pairwise comparisons selects high-performing candidates in Appendix B.6.

$$c^* = \max_c \sum_{c' \neq c} compare(c, c') \qquad compare(c, c') = \begin{cases} 1 & \text{if "prefer } c \text{ "} \\ 0.5 & \text{if "tie"} \\ 0 & \text{if "prefer } c' \text{ "} \end{cases} \qquad (1)$$

## 4 DEMONSTRATIONS

**Explore** In the *Explore* step, we generate three normal adviceand one unorthodox advice. We observe that each advice takes into account the young man's true impression, and attempts to bridge the gap between his impression and the date's expectation. Generally, we find that GPT-4 avoids the use of lies, which we believe is an important value when improving interpersonal communication.

The unorthodox advice provides a unique and unexpected strategy for the young man, and is natural to implement. In general, we found that unorthodox advice are both reasonable and creative, surpassing our expectations and demonstrating the value that EGS can bring into everyday situations.

**Generate** In the *Generate* step, we have two candidates each generated with two pieces of advice. The first candidate uses normal advice 1 in "brings out the sparkle in your eyes", and the unorthodox advice in "I can see how happy it makes you feel". In general, GPT-4 often adapts or embellishes advice, such as adjusting "color" to "sparkle" to presumably account for the black color of the jacket.

The second candidate interleaves the two advice. It starts off with "That's an edgy look for you" from advice 2, moves to "can be a bit tricky to pull off" from advice 1, and returns with "I'm having a blast seeing you try out new styles". Generally, we observe that the *Generate* step is flexible in how it incorporates multiple advice, seamlessly connecting them with each other and with the scenario.

**Simulate** We provide a simulated comparison between the two candidates shown. For Scenario 1, it starts with a summary: "straightforward compliment, focusing on the effect the jacket has on [...] my eyes, and [...] my happiness". Then, it moves to what it means to them: "He acknowledges my feeling of joy". Lastly, this forms reflection about him: "It shows he's paying close attention to me not only at visual but also at emotional level". Hierarchical reasoning is common in the analyses, allowing the simulated comparisons to consider many facets of the communication candidates.

Table 1: EGS with GPT-4 on the "White Lie During Date" scenario. The **Simulate** step uses the date's perspective to compare between the two candidates shown. In the comparison, the left candidate in the **Generate** step is Scenario 1 and the right candidate is Scenario 2.

| **Scenario: White Lie During Date** |
| --- |
| A teenage guy is currently out on a date with a girl at the mall. He is attracted to the girl and wants to make the date as enjoyable for her as possible. They are currently in a clothing store and he is waiting while his date is trying out an outfit in the changing room. She comes out of the changing room looking very pleased while wearing a black denim jacket, strikes a pose, and asks, "what do you think!". Secretly, he thinks the jacket doesn't look good on her at all, but decides he should try to compliment it anyway. Right now, he is wondering what to say. |

**Explore**

| **Normal advice 1** | **Normal advice 2** | **Normal advice 3** | **Unorthodox advice** |
| --- | --- | --- | --- |
| Share an affirmation that re-establishes your interest in her, such as "That style can be challenging to pull off, but you certainly can make anything look good." | Provide a suggestive compliment on the jacket, for instance, "Wow! That's an edgy look for you. It's fun seeing you try out new styles." | Compliment elements surrounding the jacket, like saying, "That jacket really brings out the color in your eyes," so you avoid commenting on the jacket's overall look yet keep the comment positive and about her. | Compliment the joy the jacket brings her, rather than the jacket itself. |

**Generate**

| **Normal advice 3 + Unorthodox advice** | **Normal advice 1 + Normal advice 2** |
| --- | --- |
| Wow, I love how that jacket brings out the sparkle in your eyes, and I can see how happy it makes you feel. It certainly adds a unique flair to your whole look. | Wow! That's an edgy look for you, it really shows off your daring side. You know, that style can be a bit tricky to pull off, but it seems like there's nothing you can't make look good. I'm having a blast seeing you try out new styles. |

**Simulate**

The compliment and feedback I prefer will largely depend on my personality and what I value from my date's response.

In Scenario 1, he gave a straightforward compliment, focusing on the effect the jacket has on my physical appearance, particularly my eyes, and how it elevifies my happiness. He acknowledges my feeling of joy and captures the personalized aspect of my style, which I really appreciate. It shows he's paying close attention to me not only at visual but also at emotional level.

In Scenario 2, he described how the jacket contributes to my overall style, suggesting I'm daring and edgy. He admires how I pull off tricky styles, which indicates he thinks I have a good sense of fashion. However, the phrase "it seems like there's nothing you can't make look good" might come off as a generic compliment. But the excitement he's showing in wanting to continue watching me try new styles implies he's also interested in sharing the shopping experience together.

Considering all points, while both scenarios provide complimentary feedback, I prefer Scenario 1. It resonates well as he not only notices the physical aspects but also the happiness it brings me, which very well shows he cares about my feelings.

**Answer:** Scenario 1

For Scenario 2, the simulated reaction judges "there's nothing you can't make look good" as generic, which is an interpretation that is very reasonable but also not universal. In cases like this where the audience is underspecified and their reaction is unpredictable, we find that the LLM often adopts a reaction that represents a majority of the audience demographic. We believe this is a large part of why *Simulate* is able to achieve a high agreement with human raters and online users.

## 5 HUMAN EVALUATIONS

### 5.1 DATA COLLECTION

For each scenario, we collect human ratings on all candidates from the *Generate* step, two baselines (GPT-4 zero-shot and zero-shot CoT), and an ablation altering the search space of the *Explore* step.

In EGS, we use three normal and one unorthodox advice. In a pilot, we found that 3 pieces of advice performed worse than using 1 or 2 (normalized scores 0.47 vs. 0.55 and 0.56), and so we limit advice sets to two pieces of advice max (see Appendix B.9), yielding 10 advice sets. Following Liu & Shah (2023), we generate three candidates for each set for a total of 30 candidates per scenario.

**Baselines.** In GPT-4 zero-shot, we provide the scenario, communicator, and action and generate an utterance directly. In GPT-4 zero-shot CoT (Wei et al., 2022), we ask the model to reason about

Table 2: The best candidate message selected by EGS outperforms GPT-4 zero-shot in human ratings across all constructed scenarios, and outperforms GPT-4 with CoT in five scenarios and a subset of a sixth scenario. *, **, and *** denote $p < 0.05$, $p < 0.01$, and $p < 0.001$ when compared to EGS.

| Scenario | GPT-4 zero-shot | Chain-of-Thought | EGS (ours) |
|---|---|---|---|
| Plane Crash | 6.83** | 5.98*** | **7.95** |
| Product Launch | 5.73** | 5.95* | **7.05** |
| Bargaining | 4.68 | **5.98** | 5.85 |
| Barista | 5.58 | **5.78** | 5.40 |
| Sharing Secrets | 3.67*** | 4.17** | **5.55** |
| Dating App | 5.42 | **6.48**** | 5.05 |
| White Lie During Date | 6.12 | 6.02 | **6.70** |
| Marriage Argument | 6.78* | 6.70* | **7.80** |
| **Average** | 5.60*** | 5.88** | **6.42** |

the scenario before providing what it would say (see Appendix A.7.2). We provide justification for why we select these baselines in Appendix A.7.1. For each baseline, we generate three candidates. For the *Explore* ablation, we prompt the LLM to generate advice that are encouraging or irrelevant rather than conceptual. We use three pieces of advice each, resulting in 18 candidates per scenario.

**Ratings.** Human ratings were on a 0-10 Likert scale, with (0) highly negative, (5) relatively neutral, and (10) highly positive impact on the communicator's goal. For the scenario in (Section 4):

(0) "I think his comment would make me enjoy the date a lot less."
(5) "I think his comment would not affect how much I am enjoying the date."
(10) "I think his comment would make me enjoy the date a lot more."

**Participants.** Our dataset comprises 12180 human judgments from $N = 652$ UK participants crowdsourced via Prolific. All participants provided informed consent prior to participation in accordance with an approved institutional review board protocol, and were paid 12 USD per hour. We collected 20 judgments per candidate and baseline, and each participant provided $\leq 20$ judgments. This yielded an excellent average inter-rater reliability of $r = .82$. More details are in Appendix A.8.

## 5.2 RESULT: EGS OUTPERFORMS GPT-4 ZERO-SHOT AND CoT

Averaging across all scenarios, the average human ratings of EGS outperforms GPT-4 zero-shot by 0.82 (14.6%) and CoT by 0.54 (9.2%), both statistically significant at $\alpha = 0.01$ using bootstrapping with 10000 samples. Separating by scenario, EGS outperforms GPT-4 zero-shot in all scenarios, and CoT in five scenarios (Table 2), of which four each are statistically significant at $\alpha = 0.05$.

All outputs from EGS surpassed a mean score of 5, indicating that they all had a positive impact on the communicator's goal, whereas this was not the case for either baseline. In the Bargaining scenario, we find a large discrepancy between human and GPT-4 preferences on the unorthodox advice (see Appendix B.1 for investigation). After reducing the *Explore* space by removing the unorthodox advice, EGS outperforms CoT by a large margin ($5.85 \rightarrow 6.60$ vs. 5.98).

## 5.3 RESULT: SIGNIFICANT AGREEMENT BETWEEN HUMAN SCORES & GPT-4 COMPARISONS

**Multilevel model across scenarios.** Using a multilevel model, we analyze the agreement between GPT-4 and human raters by assigning scenario as a random effect. We measure if candidates preferred in pairwise comparisons by the combined stakeholders had higher mean scores from human raters than those less preferred. For more details, please see Appendix B.2.

The results revealed a significant fixed effect for the pairwise judgements on the score provided by human raters (coef $= 0.427, p = 0.041$), demonstrating significant agreement between GPT-4 and human scores across scenarios. This effect differed across scenarios, indicating the multilevel model's appropriateness in taking into account the hierarchical nature of our data.

Table 3: We find significant agreement across GPT-4 and human ratings in five scenarios and a modified sixth scenario. Preferred and less preferred values are mean scores across all GPT pairwise evaluations, with standard errors of the mean. *, **, and *** denote $p < 0.05, 0.01$ and $0.001$ respectively. Agreement is a percentage agreement across pairs.

| Scenario | Preferred | Less Preferred | Agreement |
|---|---|---|---|
| Plane Crash | **6.19 ± 0.03*** | 5.86 ± 0.03 | 0.63 |
| Product Launch | **6.20 ± 0.03*** | 5.87 ± 0.03 | 0.67 |
| Bargaining | 5.90 ± 0.03 | **5.99 ± 0.02*** | 0.53 |
| Bargaining (-unorthodox advice) | **6.35 ± 0.04*** | 6.06 ± 0.03 | 0.69 |
| Barista | **4.66 ± 0.08*** | 3.53 ± 0.09 | 0.64 |
| Sharing Secrets | **5.72 ± 0.03*** | 4.99 ± 0.04 | 0.78 |
| Dating App | 5.24 ± 0.03 | **5.44 ± 0.03*** | 0.41 |
| White Lie During Date | 6.70 ± 0.03 | **6.81 ± 0.03** | 0.43 |
| Marriage Argument | **6.34 ± 0.04*** | 6.01 ± 0.03 | 0.65 |

**GPT-4 comparisons vs. human ratings per scenario.** For each scenario, we conducted a paired samples t-test across the preferred and less preferred candidates of the pairwise comparisons, and find that the preferred have significantly higher scores in 5/8 scenarios with $\alpha = 0.001$ (see Table 3).

Though this metric allows us to perform statistical tests, it is largely affected by easier comparisons which have a large disparity in scores, i.e., comparisons where one candidate is clearly better than the other. Thus, we follow with a percentage agreement analysis where each pair is weighted equally.

**Percentage agreement within individual scenarios.** For each pair of candidates, we aggregate the pairwise comparisons made by audiences using a weighted sum, and compare the outcome with the mean human scores of each candidate to see if they match. Tied comparisons are labeled as a half-match. We divide the matching pairs by the total pairs to obtain percentage agreement. For a mathematical formulation and justification of why we choose this metric, please refer to Appendix B.4.

In five scenarios and the modified bargaining scenario, we find agreement>0.6 between human raters and GPT-4 (Table 3). In Appendix B.5, we do the same analysis with only non-borderline cases, and find the agreement increases accordingly, with four scenarios eventually reaching 0.8 agreement with moderate sample size with a high borderline threshold.

## 6 BROADER INTERNET USER SIMULATION

We further evaluate EGS's audience simulation component on a broader space of interaction using the Stanford Human Preferences (SHP; Ethayarajh et al. 2022) dataset. SHP contains 385K human preferences over responses to online forum posts across a wide variety of subject areas from cooking to legal advice, making it a robust test bed for the simulation of different audiences. Each entry contains a forum post, two comments from the discussion, and the number of upvotes each comment received from forum users. We provide two examples of the questions/scenarios in Tables 16 and 17.

We evaluate three methods on their accuracy of predicting the comment with more upvotes. First, we use a CoT baseline, which prompts the LLM to reason about the post before predicting which comment has more upvotes. Next, we use two versions of *Simulate*, each with an audience that we define beforehand. EGS Redditor simulation (Default) takes the perspective of a Redditor browsing the forum before prompting it to reason about which comment it is more likely to upvote; EGS Redditor simulation (Funny) additionally specifies that the simulated Redditor is more likely to upvote funny and entertaining comments. Prompts and output examples can be found in Appendix C.1.

Following the authors, we filter SHP data by a ratio threshold of 3, ensuring that the more preferred comment is nontrivially preferred in each pair. To reduce the cost of API access, we select 5 subreddits and randomly sample 100 test examples from each, resulting in a total of 500 evaluations.

We observe that EGS Redditor simulation is equal or better than the CoT baseline (Table 4), suggesting that the LLM is able to make decisions more aligned with real users when explicitly prompted to simulate them. Directing the model to look for funny/entertaining comments can also significantly boost performance on more casual forums such as cooking and legal advice.

Table 4: EGS conditioned on different audience prompts outperforms GPT-4 w/ CoT on user preferences. Legaladvice and askculinary are more casual, where a fun-seeking audience leads to higher performance, while default audience is more accurate on serious forums such as asksocialscience.

| Domain | Chain-of-Thought | EGS Redditor Sim. (Default) | EGS Redditor Sim. (Funny) |
|---|---|---|---|
| legaladvice | 71.0 | 70.0 | **76.0** |
| askculinary | 59.0 | 60.0 | **70.5** |
| askhr | 72.0 | **76.0** | 72.5 |
| eli5 | 74.0 | **76.0** | 71.5 |
| asksocialscience | 77.8 | **79.4** | 61.9 |

In domains such as asksocialscience where strict rules are enforced on the informativeness and sincerity of comments, the demographic of viewers match Redditor Simulation (Default) more, and the performance of EGS Redditor simulation (Funny) drops accordingly. We perform a more cohesive investigation into redditor personalities and how they affect performance in Appendix C.2. Our results further validate that the *Simulate* component can generalize to diverse scenarios, and demonstrates that its performance can be further boosted with a better understanding of the audience.

## 7 DISCUSSION

We discuss benefits and limitations of EGS. In a more detailed discussion (Appendix E), we highlight extensions to user controls and multi-turn conversations, applications in counterfactual reasoning and human studies + RLHF, and the meta-level concepts of optimal simulation granularity, viewing LLMs as shared cultural experience, and broader impacts.

### 7.1 INTERPRETABLE EXPLANATIONS AND IMMEDIATELY AVAILABLE ALTERNATIVES

A benefit of EGS is that it provides easily accessible and interpretable explanations. Each comparison between candidates contains detailed reasoning, so users can easily find explanations for why a candidate is preferred over another from the perspective of any audience. Combined, these form a collective explanation for how the framework chose the best candidate and advice. If a user dislikes EGS's suggestion, they can access a list of alternatives that also did well in *Simulate*, or change the stakeholder weights and aggregation mechanism (Appendix B.13) based on their preferences.

### 7.2 SCALABLE EPISODIC FUTURE THINKING

A key contribution of EGS is as a scalable alternative to episodic future thinking. In our experiments, each audience in each scenario performed 1305 pairwise comparisons. This took between 2-4 hours with one API key in Sep. 2023 (10000 tokens/min), averaging to one simulated comparison every $6$-$11s$. While human simulations of the future are limited by the linear stream of consciousness over time, EGS simulations can be parallelized to achieve speeds much faster than human reasoning.

### 7.3 LIMITATIONS AND POTENTIAL NEGATIVE USE

We acknowledge that EGS is dual-use and can be used to optimize communications detrimental to society. While EGS can simulate readers to improve emails, it can also increase the success of phishing. Though GPT-4 exhibits desirable qualities like avoiding lies, other LLMs may not meet the same standards, so EGS may select utterances that superficially improve communication through deceit or manipulation, or hallucinate personal details that do not exist (see Appendix B.15 for analysis). As more models are trained using RLHF to recognize and refuse queries with malicious intent (Touvron et al., 2023; Huang et al., 2023), the safety concerns of EGS will also improve.

We also acknowledge that our scenarios fulfill common social roles, e.g., a young man struggling to compliment his female date. We encourage future work to analyze simulated audiences with more diverse backgrounds, as being less represented in training data may affect their simulation accuracy. EGS may also adopt inherent weaknesses of LLMs including social biases that may seep into decision making. Thus, we recommend users to validate outputs before putting them into use.

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

# A  Scenarios, prompting pipeline, and data

## A.1  Ten Fundamental Processes in Interpersonal Communication

In this section, we provide more in-depth descriptions of the ten fundamental processes of interpersonal communication (Berger & Roloff, 2019). These ten fundamental processes are seen as processes that underlie social interaction, and can be present regardless of the social context of the communication itself.

- Social Influence: When people's opinions or behavior are not in alignment, and they try to influence each other.

- Social Support: To effectively support those experiencing distress.

- Relationship Development: Separated into development (including initiation), maintenance, and deterioration (disengagement) of relationships.

- Deception: Includes how performing deception may alter behavior, and the degree to which individuals are skilled at identifying deception. White lies fall under this category.

- Bargaining and Negotiation: As exchanges involve risk, especially when partners are unsure that they can trust each other, communication processes have evolved to ensure fair exchanges.

- Conflict Management: Managing the negative consequences of conflict and how individuals confront each other.

- Conversation Management: This suggests that individuals in a conversation implicitly understand that each will contribute to the conversation and advance its point.

- Impression Management: Individuals form impressions of each other that can influence both the courses of their interactions as well as future interactions and decisions.

- Privacy Management: This involves the causes and consequences of self-disclosure, which is the tendency for individuals to reveal personal information to others.

- Uncertainty Management: As individuals cannot be completely certain of their conversational partners' current emotional states, beliefs, attitude, and future actions, they engage in social interaction under conditions of uncertainty.

Table 5: The coverage between the ten fundamental processes of interpersonal communication and our constructed scenarios.

| | Social Influence | Social Support | Relation-ship Dev. | Deception | Bargain & Neg. | Conflict Mgmt. | Conversation Mgmt. | Impression Mgmt. | Privacy Mgmt. | Uncertainty Mgmt. |
|---|---|---|---|---|---|---|---|---|---|---|
| Plane Crash | ✓ | ✓ | | | | | | ✓ | | |
| Product Launch | ✓ | | | | ✓ | | | ✓ | | |
| Bargaining | ✓ | | | | ✓ | | | | | ✓ |
| Barista | ✓ | | | | | | ✓ | | | ✓ |
| Sharing Secrets | | | ✓ | | | | ✓ | ✓ | ✓ | |
| Dating App | | | ✓ | | | | | ✓ | | ✓ |
| White Lie During Date | | | ✓ | ✓ | | | | | ✓ | |
| Marriage Argument | ✓ | | ✓ | | | ✓ | | | | |

## A.2 SCENARIO CHOICES AND DETAILS

In this section, we provide the descriptions for the eight scenarios we construct to evaluate our framework, followed by the fundamental processes they cover and their interesting qualities. Table 5 provides a visual map for the coverage between fundamental processes and scenarios. The first four scenarios focus on more formal relationships, while the latter four are from more personal relationships. The scenarios also span various venues of communication, from speaking to written to digital. In constructing these scenarios, we hope to cover a wide range of potential situations that our framework could help assist people in.

**Plane Crash**  An airline spokesperson wants to minimize the negative public opinion of their company. Earlier today, a flight of the airline crashed due to unknown causes and there were 2 passengers that died. Right now, they are about to make the first public statement to the media at a live press conference. They only have 30 seconds of airtime to speak about the issue.

The scenario focuses on the fundamental process of social support, while also covering impression management and social influence. A key objective in minimizing the negative public opinion is to provide support to the families and friends of the victims that are in experiencing distress. It also embodies a case where experimenting with the message is too risky for the agent and their company. Thus, within the large area of unexplored possibilities, there may be a better message to communicate available. Through the simulation framework we are able to ignore the risks and arrive at a reasonable outcome.

**Product Launch**  A company CEO is announcing their new smartphone product, the jPhone, and they want to maximize its sales. The jPhone has the following features:
- Processing speeds equal to the top smartphones on the market
- Priced at $700, which is 10-30% cheaper than competitor smartphones on the market
- A unique, innovative design and user interface
Right now, they are about to give a quick 30-second presentation about the jPhone, broadcasted to major television channels.

This scenario focuses on the fundamental process of social influence and impression management, but also covers the seller's facet in the bargaining and negotiation process, as one of the generated pieces of advice includes offering a flash sale for the first customers to purchase the jPhone.

**Bargaining**  Jill wants to buy a round flower vase at a local artisan market. The marked price for the vase is 30 dollars. Jill would be willing to buy the vase at its original price, but she would also like to spend as little money as possible while still purchasing the vase. Right now, she is about to speak to the artisan selling the vase.

This scenario focuses on the fundamental process of bargaining and negotiation, while also covering social influence and uncertainty management, as Jill does not know much the artisan is willing to lower their price. Together with the product launch scenario, we consider communication optimization from both seller and buyer perspectives.

**Barista** A barista at a coffee shop wants to maximize their tips received from customers. During their shift, a customer walks to the counter and says, "Hello, can I have a latte with whole milk, medium size?". The barista has a few seconds to respond.

This scenario primarily covers the social influence fundamental process, while also including uncertainty management and conversation management. Customers arriving at a coffee shop may provide surface-level cues as to how willing they are to tip relative to their norm, but there is still uncertainty to what they would typically tip. Furthermore, this scenario embodies a case where the best conceptual knowledge may not be widely shared across geographical or cultural distances, but has a potential to improve the quality of life of both customers and baristas alike.

**Sharing Secrets** Mary is acquaintances with Carla through her job. She is currently chatting with Carla after work, and she wants to become closer friends with Carla. She thinks sharing a secret of hers might bring them closer together. The secrets that she can think of sharing, from least to most personal, are:
1. She used to hate pickles as a child but they have grown on her.
2. She has recently been feeling unconfident about her ability to succeed in her job.
3. She used to be impulsive when spending money and still has some debt.
Right now, she feels like it is a good point in the conversation to share a secret. She decides to try and do so.

This scenario primarily focuses on the privacy management fundamental process, but also covers relationship development, conversation management, and impression management. This scenario differs from the others as there are a few options for the model to decide between, which aligns with the goal of being authentic when developing closer relationships. Furthermore, a selection paradigm is different from the original generation paradigm as the solution space is much more restricted, allowing for different qualities of the framework to be evaluated.

**Dating App** A young man named Eric is in his early 20s and wants to find a girlfriend. He is currently an undergraduate student at a large state university, and likes to play tennis and hang out with friends in his free time. He likes animals and has a beagle dog named Scott. He would rate his looks and height at about average compared to those around him. He has decided to try his luck on a dating app. Right now, he is drafting his profile bio, and he is wondering what to write.

This scenario primarily focuses on impression management and uncertainty management, with a smaller emphasis on the initiation stage of relationship development. On a dating app, who views your profile is unknown to you beforehand, and potential matches may vary widely in personality and what they prefer. Thus, the framework would need to take into account this uncertainty when optimizing for the best communication.

**White Lie During Date** A teenage guy is currently out on a date with a girl at the mall. He is attracted to the girl and wants to make the date as enjoyable for her as possible. They are currently in a clothing store and he is waiting while his date is trying out an outfit in the changing room. She comes out of the changing room looking very pleased while wearing an black denim jacket, strikes a pose, and asks, "what do you think!". Secretly, he thinks the jacket doesn't look good on her at all, but decides he should try to compliment it anyway. Right now, he is wondering what to say.

This scenario primarily focuses on the deception fundamental process, while also covering the relationship development and privacy management processes. Deception is an important part of our lives, and sometimes we need to lie about our impressions in order to achieve our goals. We know that state-of-the-art LLMs can lie through hallucinations, but to our best knowledge lying intentionally has yet to be studied.

**Marriage Argument** A woman is currently upset with her husband about the fact that he is consistently staying out with his friends until late at night instead of coming home to spend time with

her. However, her husband is quite stubborn, and does not like to admit that he is in the wrong. She knows that her husband would be able to see reason eventually, but wants to minimize the time that it takes for him to see reason. Right now, she is about to approach him to talk about the issue.

This scenario primarily focuses on the conflict management, while also including the maintenance aspect of the relationship development process and the social influence process. In delicate situations such as these, simulations can be very helpful in providing feedback and suggestions to the user.

## A.3 ADVICE GENERATION PIPELINE AND EXAMPLES

In this section, we describe the pipeline and prompts used to generate the advice used in the *Explore* step, and showcase the generated advice for each scenario.

### A.3.1 ADVICE GENERATION PIPELINE

Starting from a provided scenario, we append the following questions in order to search for conceptual normal advice and unorthodox advice:

**Prompt:**

[Scenario]

Please give them [num_advice] different pieces of conceptual advice (each one sentence long, in a numbered list) on what they should say to help them achieve their goal.

-or-

Please give them [num_unorthodox_advice] different pieces of advice (each one sentence long, in a numbered list) that are unorthodox but potentially helpful for their task and goal.

### A.3.2 ADVICE GENERATED PER SCENARIO

**Scenario: Plane Crash**

**Normal Advice:**
Express sincere empathy and condolences towards the victims and their families to show that your company holds human life in the highest regard.

Regretfully acknowledge the incident, ensuring that the necessary investigations are being conducted to understand and correct the issue promptly.

Reassure the public that this is an isolated event and emphasize on your airline's long-standing reputation for safety and commitment to continuous improvement.

**Unorthodox Advice:**
Humanize the situation by expressing sincere condolences directly addressing the families affected before discussing the flight itself.

**Scenario: Product Launch**

**Normal Advice:**
Highlight the fast processing speed: "With processing speeds to match the best in the market today, the jPhone promises a seamless, lag-free user experience that makes multitasking a breeze."

Underline the competitive pricing: "Priced at just $700, the jPhone offers top of the line features at a more accessible price tag, without compromising on quality or performance."

Emphasize on the unique design and interface: "Featuring an innovative design and user interface, the jPhone introduces a new level of interactivity, transforming how you communicate, work and play."

**Unorthodox Advice:**

In your 30-second presentation, declare a limited time 24-hour sale where the first 100 customers get the phone at an additional 50% off, creating hype and urgency to buy immediately.

**Scenario: Bargaining**

**Normal Advice:**

Jill should compliment the artisan's craftsmanship while gently asking if there is any room for bargaining.

She should emphasize her interest in supporting local artisans but also give a hint that the price is a bit higher than what she can afford.

She should ask if there are any discounts available for purchasing multiple items or returning customers as she plans to come back to the market in the future.

**Unorthodox Advice:**

Jill, consider offering to promote the artisan's work on your social media in exchange for a discount on the vase.

**Scenario: Barista**

**Normal Advice:**

Always respond with a friendly demeanor and vocal tone, saying something like, "Of course, I'll get a medium whole milk latte ready for you right away!"

Show professionalism by demonstrating knowledge, such as offering, "Sure, that will be with our freshest whole milk, which gives the latte a really creamy texture."

Personalize the service with a comment like, "Sure, a medium whole milk latte coming up! That's one of my favorite drinks too, it's so smooth and flavourful."

**Unorthodox Advice:**

"Sure, I'll make it special for you - did you know customers say our lattes taste better if you pay in tips?"

**Scenario: Sharing Secrets**

**Normal Advice:**

Be open about your feelings, share something personal but not too deep - like your earlier dislike for pickles and how your tastes have changed.

Balance vulnerability with professionalism, sharing your anxieties about your job can show a softer side without making you seem incompetent.

Honesty about past mistakes can be very humbling, share your financial history in a way that shows growth and understanding from that experience.

**Unorthodox Advice:**

"Mary, try using humor to lighten up the conversation before diving into your secret, it might engage Carla more and make the whole interaction feel genuine and effortless."

**Scenario: Dating App**

**Normal Advice:**

Emphasize shared interests: highlighting hobbies, like tennis and love for animals, could attract like-minded individuals who also share these interests.

Be genuine: portray yourself honestly, including mention of your university pursuits and social habits, to find compatibility based on who you truly are.

Show a sense of humor: make sure to keep the profile light and humorous, as it can attract prospective matches that appreciate your sense of fun and approachability.

**Unorthodox Advice:**
Share a fun, unique story involving Scott, your beagle, to catch people's attention and offer an easy conversation starter.

**Scenario: White Lie During Date**

**Normal Advice:**
Compliment elements surrounding the jacket, like saying "That jacket really brings out the color in your eyes," so you avoid commenting on the jacket's overall look yet keep the comment positive and about her.

Share an affirmation that re-establishes your interest in her, such as "That style can be challenging to pull off, but you certainly can make anything look good."

Provide a suggestive compliment on the jacket, for instance, "Wow! That's an edgy look for you. It's fun seeing you try out new styles."

**Unorthodox Advice:**
Compliment the joy the jacket brings her, rather than the jacket itself.

**Scenario: Marriage Argument**

**Normal Advice:**
Appeal to his empathy by saying "Do you ever wonder how I feel when you're not home with me and I'm left to fend for myself late at night?"

Directly communicate her feelings by saying "I miss you and our time together, your late nights out are affecting our relationship."

Seek common ground by saying, "How about we both work on balancing our social lives with our time together?"

**Unorthodox Advice:**
Start your conversation by asking about his day and his friends' shenanigans, showing genuine interest, before gradually steering the topic to your need for his company.

## A.4 CANDIDATE GENERATION PIPELINE AND EXAMPLES

In this section, we describe the pipeline and prompts used to generate the message candidates in the *Generate* step of EGS, and showcase a subset of the generated candidates.

### A.4.1 CANDIDATE GENERATION PIPELINE

Each candidate is generated with a subset of the advice from the *Explore* step. Originally, we generate three candidates for each item in the powerset of the set of advice, but we limited this to items in the powerset containing two or less pieces of advice due to both worse performance in a pilot and scalability concerns.

For the prompts, we transform the scenario description into the second-person perspective, and then split it between a description of the speaker (system prompt) and the situation (user prompt). To the end of the user prompt, we append the "remember a few pieces of advice" framing and insert the set of advice, followed by asking the LLM what to say.

**System Prompt:** [second-person description of communicator from scenario]

**User Prompt:**

[second-person description of situation from scenario]

You remember {a piece / a few pieces} of advice:

[advice set]

You decide to focus on using {it / these pieces of advice} during your statement. What do you say? Answer:

### A.4.2    CANDIDATE GENERATION EXAMPLES

**Scenario: Plane Crash**

**Advice set:** {Normal advice 1}

Express sincere empathy and condolences towards the victims and their families to show that your company holds human life in the highest regard.

**Candidates:**

We deeply regret this terrible incident that occurred today. At our airline, we hold human life in the highest regard. Our hearts go out to the families and loved ones of those we lost. Please know we are leaving no stone unturned to determine the cause and to ensure such an incident does not recur. This incident does not reflect our enduring commitment to the safety and care of our passengers, which remains our highest priority.

Ladies and Gentlemen, we are absolutely devastated about the tragic incident that occurred today. We extend our heartfelt condolences and deepest sympathies to the families who lost their loved ones in this unfortunate event. The value of human life is paramount, and our thoughts are with all those affected by this incident. We're committed to investigating the event thoroughly, cooperating fully with aviation authorities to ensure this does not happen again. As information becomes available, we will promptly and transparently share updates. Our priority now is supporting the affected families during this incredibly difficult time. Thank you.

Good evening, it is with profound sadness that I stand before you today. Our hearts and thoughts go out to the families of the two passengers whose lives were tragically lost in today's unfortunate incident. We hold their memory in our thoughts and prioritize their families' well-being. At our company, every life is irreplaceable and of highest regard. We commit to conducting a thorough, transparent investigation and vow to take appropriate steps moving forward. Thank you all.

**Scenario: Product Launch**

**Advice set:** {Normal advice 1, Normal advice 2}

Highlight the fast processing speed: "With processing speeds to match the best in the market today, the jPhone promises a seamless, lag-free user experience that makes multitasking a breeze."

Underline the competitive pricing: "Priced at just $700, the jPhone offers top of the line features at a more accessible price tag, without compromising on quality or performance."

**Candidates:**

Welcome to the future of smartphones, welcome to the world of jPhone. Designed with an inherent innovative spirit, jPhone boasts processing speeds matching the very best in the market. Your experience is seamless; multitasking is a breeze. Chunky price tags? A problem of yesterday, because the jPhone comes at a refreshingly accessible price of just $700. However, we don't believe in compromises, thus equipping you with top of the line features without denting your pocket. Experience the unique, innovative design and interface that jPhone offers. Welcome to a smart world! Get your jPhone today.

Welcome to the revolution in smartphone technology, welcome to the jPhone. With processing speeds that go head to head with the best in the market, jPhone guarantees a smooth, lag-free experience that takes multitasking to new heights. And it doesn't stop there. Priced at a competitive $700, jPhone breaks the norms by offering top-tier features at this accessible price, without any compromise on quality or performance. Experience power, performance, and innovation quite like never before, only with jPhone!

Good evening, everyone. I'm thrilled to introduce our latest innovation, the jPhone. Known for blazing-fast processing speeds, we ensure an unparalleled, lag-free user experience that outperforms the best in the market; making even multitasking smooth and effortless. What's more exciting? We

deliver this exceptional performance at a price that won't break your bank. Only $700 for a superior smartphone experience. Furthermore, our innovative design and unique user interface will change your perspective on smartphone usage. Step into the future with the jPhone - where excellence meets affordability and innovation. Thank you!

**Scenario: White Lie During Date**

**Advice set:** {Normal advice 2, Unorthodox advice}

Share an affirmation that re-establishes your interest in her, such as "That style can be challenging to pull off, but you certainly can make anything look good."

Compliment the joy the jacket brings her, rather than the jacket itself.

**Candidates:**

Wow, I must say that style isn't easy to pull off, but you certainly can! You just light up in whatever you wear and that joy is contagious. It's not everyone who can wear a jacket like that and make it work. You make it feel like more than just a jacket, you bring it to life!

You look really happy in that jacket. It's a tough style to pull off, but you know what? You can make anything look good!

Well, that's a bold style of jacket, but honestly, you could make anything look good. It's clear that you're really happy wearing it, which is what matters. That smile of yours lights up the room even more than any outfit could.

### A.5    AUDIENCE GENERATION PIPELINE AND EXAMPLES

In this section, we describe the pipeline and prompts used to generate the potential audiences and their corresponding weights and profiles, and showcase the generated profiles and weights.

#### A.5.1    AUDIENCE GENERATION PIPELINE

Starting from a provided scenario, we query the LLM for stakeholder audiences and weights:

**Prompt:** Please reason about and provide a list of up to $x$ stakeholders that are most relevant to the main character's goal, as well as a numerical value from 1 to 100 weighting the relative importance of each stakeholder. You do not need to reach the maximum amount of stakeholders, but instead include only stakeholders that would have an influence on the main character's goal. State your answer in the following format:
1. stakeholder name (stakeholder weight): justification
2. ...

Then, once we have the stakeholders, we generate their description:

**Prompt:** Please write a description of the following role for the following scenario. It is important for the description to include a communication from the main character of the scenario. Here are a few examples for various stakeholders and scenarios:

[few shot examples]

Your task:

Scenario: [provided]

Role: [provided]

Description:

We then generate a question to ask the stakeholder that accurately reflects their influence over the communicator's goal:

**Prompt:** You are asking a question to a stakeholder to have them choose between the better of two scenarios. It is important that the question be tied directly to the goal that the main character of the story wants to achieve. Here are a few examples for various stakeholders and stories:

[few shot examples]

Your task:

Story: [provided]

Stakeholder: [provided]

Question:

The question generated is used to fill in the following template:

**Template:**

Scenario 1: [candidate 1]

Scenario 2: [candidate 2]

[insert question] Please think step by step and explain your reasoning. Put your response in the following format: [reasoning text]

Answer: Scenario [1 or 2]

Both the description and the filled template are used when querying for the simulated audience's pairwise comparison of two potential candidates. The description is used as the system prompt and the template containing the scenarios and the question as the user prompt.

A.5.2   AUDIENCE GENERATION EXAMPLES

**Scenario: Plane Crash**

**Stakeholder Audiences:** General public (weight 60), Victims' families (weight 40)

**Justification from LLM:**

General Public (60): The general public is the largest stakeholder in this context as their opinion largely controls the reputation and success of the airline. The impact of a plane crash is immense and the public reaction towards the airline's statements and actions will be a critical factor in shaping future customer viewership and company reputation. The numerical value is deliberately higher as they are many and diverse and influence others.

Victims' Families (40): The families of the victims who died in the crash are also directly connected to the incident. They need to be managed and supported well to ensure that the airline shows empathy, responsibility, and care towards its passengers. While numerically less than the general public, respecting their sentiments will also influence public opinion in a substantial manner. This is why they are weighted slightly lower but are paramount.

**Audience - General public:**

- Description (system prompt):

  You are an everyday citizen watching a live press conference on TV regarding a recent airline crash. Naturally, you are concerned about the safety of flying and are interested in hearing the airline's response to this tragic event. This event might influence your future travel decisions, especially if you usually choose to fly with this particular airline. The spokesperson for the airline is about to speak on behalf of the company.

- Template + Question (user prompt):

  Scenario 1: [Candidate 1]

  Scenario 2: [Candidate 2]

  In which scenario would you be more likely to maintain a favorable opinion of the airline and consider using their services in the future? Please think step by step and explain your reasoning.

  Put your response in the following format:

[reasoning text]

Answer: Scenario [1 or 2]

**Audience - Victims' families:**

- Description (system prompt):

  You are a family member of one of the victims in the recent airline crash. You're watching a live press conference, hoping to get more information on the incident from the airline's spokesperson. They are about to make the first public statement to the media, and their explanation and evident concern can greatly affect your feelings towards the company in this tragic circumstance.

- Template + Question (user prompt):

  Scenario 1: [Candidate 1]

  Scenario 2: [Candidate 2]

  In which scenario would you be more likely to feel empathized with and reassured about the airline's commitment to rectify the situation? Please think step by step and explain your reasoning.

  Put your response in the following format:

  [reasoning text]

  Answer: Scenario [1 or 2]

**Scenario: Product Launch**

**Stakeholder Audiences:** Potential consumers (weight 60), Company's sales and marketing team (weight 30), Media outlets (weight 10)

**Justification from LLM:**

Potential consumers (weight 60): These represent the main target market of the new smartphone product, therefore, have direct impact on the success of the jPhone. The more consumers get attracted to its features and price, the higher the sales revenue the company can generate.

Company's sales and marketing team (weight 30): It is crucial to the success of the jPhone that this team effectively markets and communicates its features and benefits to the potential consumers. Good marketing strategies will result in higher consumer interest and therefore higher sales.

Media outlets (weight 10): They do not directly impact the sales of the product, but they can influence potential consumers' opinions about the jPhone. Positive reviews and features in popular outlets can boost the product's image and therefore indirect affect sales.

**Audience - Potential consumers:**

- Description (system prompt):

  You are a potential consumer watching a brief television announcement about a newly-released smartphone, called the jPhone. You're intrigued by the latest technology and are considering buying a new phone. You are waiting to hear the CEO's 30-second presentation about this product's features, hoping for an informative snapshot that might influence your purchasing decision. The CEO is about to start their presentation.

- Template + Question (user prompt):

  Scenario 1: [Candidate 1]

  Scenario 2: [Candidate 2]

  In which scenario would you be more likely to buy the jPhone? Please think step by step and explain your reasoning.

  Put your response in the following format:

  [reasoning text]

  Answer: Scenario [1 or 2]

**Audience - Company's sales and marketing team:**

- Description (system prompt):

  You are part of the company's sales and marketing team watching the live TV broadcast of the CEO presenting the new smartphone product, the jPhone. As a team member, you are particularly interested in the key selling points and unique features of the jPhone that could help you in creating marketing strategies and campaigns aimed at maximizing its sales. You listen attentively to the CEO's 30-second product pitch, ready to incorporate his message in your sales and marketing plans.

- Template + Question (user prompt):

  Scenario 1: [Candidate 1]

  Scenario 2: [Candidate 2]

  In which scenario would the product sales campaign be most successful in maximizing jPhone sales? Please think step by step and explain your reasoning.

  Put your response in the following format:

  [reasoning text]

  Answer: Scenario [1 or 2]

**Audience - Media outlets:**

- Description (system prompt):

  You are a representative from one of the major media outlets attending the company CEO's presentation of the new smartphone product, the jPhone. Your job is to listen to the CEO's presentation, understand the key features and selling points of the smartphone, and relay the information to the public through your media platform. You are highly interested in the competitive price point and the features that differentiate this phone from other competitors on the market. The CEO is currently starting their 30-second presentation about the jPhone.

- Template + Question (user prompt):

  Scenario 1: [Candidate 1]

  Scenario 2: [Candidate 2]

  In which scenario would you be more likely to give more media coverage and promotion towards the jPhone? Please think step by step and explain your reasoning.

  Put your response in the following format:

  [reasoning text]

  Answer: Scenario [1 or 2]

**Scenario: Bargaining**

**Stakeholder Audiences:** Artisan (weight 100)

**Justification from LLM:**

Artisan (100): This is the direct seller of the vase and has full control over the price. They are the ones Jill has to negotiate with in order to potentially lower the price.

**Audience - Artisan:**

- Description (system prompt):

  You are an artisan at a local market, selling an array of handcrafted items including a round flower vase that has caught the attention of a customer named Jill. The vase is marked at 30 dollars. Jill is interested in the vase and is approaching you, possibly to negotiate the price.

- Template + Question (user prompt):

  Scenario 1: [Candidate 1]

  Scenario 2: [Candidate 2]

  In which scenario would you be more likely to negotiate on the price of the vase with Jill? Please think step by step and explain your reasoning.

  Put your response in the following format:

  [reasoning text]

  Answer: Scenario [1 or 2]

**Scenario: Barista**

**Stakeholder Audiences:** The Customer (weight 100)

**Justification from LLM:**

The Customer (100): The ultimate stakeholder in regards to maximizing tips is the customer, as they are the ones directly contributing to the tips. The service experience, drink quality, and overall satisfaction of the customer will directly impact the tip amount they leave. Therefore, the barista should focus their attention towards meeting and exceeding the customer's expectations throughout the whole interaction.

**Audience - The Customer:**

- Description (system prompt):

  You are a customer at a coffee shop ordering a drink. You approach the counter and say to the barista, "Hello, can I have a latte with whole milk, medium size?". The barista is briefly taking a pause before responding to your order.

- Template + Question (user prompt):

  Scenario 1: [Candidate 1]

  Scenario 2: [Candidate 2]

  In which scenario are you more likely to leave a tip? Or if you would leave a tip in both cases, in which scenario would you leave a larger tip? Please think step by step and explain your reasoning.

  Put your response in the following format:

  [reasoning text]

  Answer: Scenario [1 or 2]

**Scenario: Sharing Secrets**

**Stakeholder Audiences:** Carla (weight 100)

**Justification from LLM:**

Carla (100): Carla is the individual with whom Mary is trying to foster a closer relationship. Therefore, her perceptions, reactions, and feelings are extremely pertinent to Mary's goal. Moreover, her understanding and acceptance of Mary's disclosed secret would contribute towards building a more personal relationship between the two. It's also important that Mary gauges Carla's response to ensure she doesn't share something too personal prematurely, which may discomfort Carla and harm their budding friendship. Hence, Carla's interest and comfort level should be the primary consideration in making this decision.

**Audience - Carla:**

- Description (system prompt):

  You are Carla, a colleague of Mary. You're currently having a casual chat after work and are getting along well with her. There's a lull in the conversation and it appears that Mary is about to share something personal to create a deeper bond between the two of you.

- Template + Question (user prompt):

  Scenario 1: [Candidate 1]

  Scenario 2: [Candidate 2]

  In which scenario are you more likely to feel closer to Mary, or if you feel closer in both cases, in which scenario do you feel significantly closer to her? Please think step by step and explain your reasoning.

  Put your response in the following format:

  [reasoning text]

  Answer: Scenario [1 or 2]

**Scenario: Dating App**

**Stakeholder Audiences:** Potential Date (weight 70), Eric himself (weight 30)

**Justification from LLM:**

Potential Date (70): The potential date is the most critical stakeholder since Eric is looking to find a girlfriend. He needs to craft a profile that will appeal to women who have similar interests and are looking for a man like him. This person's opinion of him will directly determine whether he can achieve his goal.

Eric himself (30): While the potential date's view is critical, Eric's opinion of himself is also very important. His bio needs to reflect his personality and interests accurately, so it represents him truthfully to potential dates. If he's not comfortable with what he's written or if it doesn't reflect who he is, he may not feel genuine when interacting with potential dates, which could lead to less successful dating experiences. His level of satisfaction with his bio is important for his overall dating app experience.

**Audience - Potential Date:**

- Description (system prompt):

  You are a young woman in your early 20s using a dating app, open to the possibility of forming a romantic relationship. You come across Eric's profile, an undergraduate student who likes tennis, hanging out with friends, and has a soft spot for animals. You are reading his bio and looking at his profile pictures, deciding whether to swipe right or left.

- Template + Question (user prompt):

  Scenario 1: [Candidate 1]

  Scenario 2: [Candidate 2]

  In which scenario are you more likely to swipe right and potentially match with Eric? Please think step by step and explain your reasoning.

  Put your response in the following format:

  [reasoning text]

  Answer: Scenario [1 or 2]

**Audience - Eric himself:**

- Description (system prompt):

  You are a young man named Eric in your early twenties looking to start dating. You are currently an undergraduate student at a large state university, you enjoy playing tennis and spending time with friends. You love animals and own a beagle dog named Scott. You believe that physically, you are about average looking and height. You have decided to try your luck at finding a girlfriend through a dating app. As you sit to draft your profile bio, you ponder on what exactly to say about yourself.

- Template + Question (user prompt):

  Scenario 1: [Candidate 1]

  Scenario 2: [Candidate 2]

  In which scenario do you think you'd be more likely to attract a potential match who shares your interests and values? Please think step by step and explain your reasoning.

  Put your response in the following format:

  [reasoning text]

  Answer: Scenario [1 or 2]

**Scenario: White Lie During Date**

**Stakeholder Audiences:** His date (weight 100)

**Justification from LLM:**

His date (100): His main goal is to make the date enjoyable for her, thus making her feelings and reactions the most critical to him at the moment. It's essential for him to respond in a way that maintains her joy and keeps the momentum of their date flowing positively. Her perception of his reaction impacts the overall success of the date, making her the utmost relevant stakeholder.

**Audience - His date:**

- Description (system prompt):

  You are a teenage girl on a date at the mall with a guy you find interesting. You have just come out of a clothing store's changing room wearing a black denim jacket you really like. Excited by the look, you turn to your date, strike a confident pose, and ask him, "what do you think!". You're eagerly waiting for his compliment or feedback on your chosen outfit.

- Template + Question (user prompt):

  Scenario 1: [Candidate 1]

  Scenario 2: [Candidate 2]

  In which scenario would you appreciate the guy's comment more or have a more enjoyable experience on the date? Please think step by step and explain your reasoning.

  Put your response in the following format:

  [reasoning text]

  Answer: Scenario [1 or 2]

**Scenario: Marriage Argument**

**Stakeholder Audiences:** Husband (weight 100)

**Justification from LLM:**

Husband (100): The husband is the only stakeholder here as his actions directly affect the issue at hand. The wife's contentment is directly tied to his willingness to change behavior and accommodate her needs. For the wife's goal to be reached, indeed, the husband must agree and commit to the changes. Therefore, his weight is 100 as he holds the entirety of the decision-making power in this scenario.

**Audience - Husband:**

- Description (system prompt):

  You are a husband who enjoys spending late nights out with your friends. Your wife is currently upset with you for the lack of time you're spending at home with her. You are known for being stubborn and not readily admitting your faults. You know your wife is about to approach you to discuss your late nights and you anticipate a challenging conversation about it.

- Template + Question (user prompt):

  Scenario 1: [Candidate 1]

  Scenario 2: [Candidate 2]

  In which scenario would you be more likely to understand and acknowledge your wife's feelings, and alter your behavior to come home earlier? Please think step by step and explain your reasoning.

  Put your response in the following format:

  [reasoning text]

  Answer: Scenario [1 or 2]

A.6    PAIRWISE COMPARISONS

We conduct the pairwise comparisons for each simulated audience using the **Template** listed above at the end of Appendix A.5.1. Each of the two candidates in the pairwise comparison is filled into Candidate 1 and Candidate 2, with their order being determined randomly to avoid positional biases in the LLM's judgements.

The output of the simulated pairwise judgement is parsed into one of {"prefer scenario 1", "prefer scenario 2", "tie"}. "Tie" is reserved for when the response insists that the choice depends on the audience's personal preference, or when the LLM is completely balanced between the two choices. In cases where the output is not in the proper answer format, we ask the LLM to re-classify the output into one of the pairwise comparison outcomes using another query containing just the answer.

## A.7 Baselines for Human Data Experiments

In this section, we describe the reasoning and prompts used for the selection and generation of baselines for the human data experiments (Section 5).

### A.7.1 Baseline Choice

In our human experiments, we select GPT-4 zero-shot and zero-shot CoT (Wei et al., 2022) as our two main baselines.

We forgo the use of any few-shot baselines due to the difficulty of identifying exemplar utterances in interpersonal communication, and the limited relevance any particular exemplar would have to the entire problem domain. Furthermore, we do not include comparisons to self-consistency methods (Wang et al., 2023b; Jain et al., 2023) due to their assumptions of a best underlying fixed answer that corresponds to the highest probability. In an ablation (Section B.8), we show that unorthodox advice contributes meaningfully towards the generation of the best-performing candidates, providing evidence that their assumption does not hold across our problem domain.

### A.7.2 Baseline Prompts

**GPT-4 zero-shot prompt:**

[Scenario description]

What would you say if you were [communicator]?

**GPT-4 zero-shot CoT prompt:**

[Scenario description]

What would you say if you were [communicator]? Please reason about the situation before providing your answer. Provide your answer in this format:

Reasoning: [text]

Answer: [text]

We generate three responses for each baseline, and each response is rated by 20 human participants on a 0-10 likert scale.

## A.8 Data Collection Procedure

Participants were crowdsourced through Prolific[1]. After providing an informed consent, participants received the following instructions:

> "Thank you for participating in our study! In this study, we will provide you with a series of scenarios, where in each scenario an individual will make a statement related to the scenario. Your task is to evaluate the quality of that statement with respect to the given scenario. You will have eleven response options, ranging from 0 (worst) to 10 (best). Choose the one you think is most appropriate. Note: no prior expertise is required to complete this task, just choose what you intuitively think is the right answer."

Participants then proceeded to the main experiment where they provided response-quality judgments on a 0-10 Likert scale for up to 20 randomly assigned responses. We detail the evaluation questions for each scenario below.

- Plane Crash: "Based on the spokesperson's response, how do you think the company's public opinion would change?".
- Product Launch: "If you were considering getting a new phone, how likely are you to buy a jPhone following the CEO's presentation?".

---

[1] https://www.prolific.co

Table 6: Our human ratings have high inter-rater reliability (IRR) across all scenarios. For each scenario, we provide the number of candidate messages evaluated, the number of crowdworkers recruited, and IRR with 95% confidence intervals.

| Scenario | # Candidates | # Participants | IRR |
|---|---|---|---|
| Plane Crash | 72 | 76 | $0.96 \pm 0.01$ |
| Product Launch | 72 | 78 | $0.75 \pm 0.08$ |
| Bargaining | 72 | 78 | $0.82 \pm 0.06$ |
| Barista | 102 | 103 | $0.90 \pm 0.03$ |
| Sharing Secrets | 72 | 79 | $0.77 \pm 0.07$ |
| Dating App | 75 | 80 | $0.66 \pm 0.10$ |
| White Lie During Date | 72 | 80 | $0.90 \pm 0.03$ |
| Marriage Argument | 72 | 78 | $0.77 \pm 0.08$ |

- Bargaining: "Based on Jill's response, how likely would you be willing to negotiate the price of the vase with Jill?".

- Barista: "Based on the barista's response, how likely are you to tip them?".

- Getting Close by Sharing Secrets: "As Carla, how much closer would you feel you are with Mary after what she shared?".

- Dating App: "Based on Eric's decision, how likely do you think his profile would attract a potential girlfriend?".

- White Lie During Date: "As the girl, how do you think the guy's response would affect how much you are enjoying the date?".

- Marriage Argument: "As the husband, how likely are you to admit that you are wrong immediately, instead of potentially defending yourself first or being dissatisfied at your wife's comments?".

Altogether, the number of participants, candidates, and the inter-rater reliability (IRR) computer using split-half correlation with Spearman-Brown correlation (Brown, 1910) for each scenario is provided in Table 6. We observe a high IRR across all scenarios, pointing to the quality in the human data collected.

## B   ANALYSES ON HUMAN EXPERIMENT

### B.1   BARGAINING SCENARIO'S UNORTHODOX ADVICE

When computing the analysis for the GPT-4 pairwise comparison results vs. mean human ratings per scenario (Section 5.3), we found that the Bargaining scenario contained a piece of controversial advice that was in both EGS's best advice set and best candidate, but was given a mean human rating of 5.56 (candidates without this advice averaged a score of 6.20, t-test with/without this advice $p < 0.001$):

> "Consider offering to promote the artisan's work on your social media in exchange for a discount on the vase."

When we remove this advice and re-run the framework, the difference returns to being highly significant in favor of the preferred candidate in the pairwise comparisons. For our experiments, this version of shown under "Bargaining (-unorthodox advice)".

More generally, for candidates containing unorthodox advice, not only do we find higher disagreement between human raters and EGS, we also find higher disagreement within human raters themselves in seven of the eight scenarios (see Appendix B.3). This reflects that simulating audience reactions in the less common situations created by unorthodox advice is less consistent for humans as well as LLMs.

Table 7: Inter-rater reliability (IRR) of candidates containing unorthodox advice is lower than any other subgroup. Baselines denotes IRR for GPT-4 zero-shot, GPT-4 Chain-of-Thought, and irrelevant and conceptual search space candidates. Normal Only denotes candidates without unorthodox advice in their advice sets.

| Scenario | Baselines | Normal Only | Unorthodox |
|---|---|---|---|
| Plane Crash | $0.96 \pm 0.01$ | $0.71 \pm 0.10$ | $0.70 \pm 0.13$ |
| Product Launch | $0.78 \pm 0.05$ | $0.21 \pm 0.26$ | $-0.40 \pm 0.59$ |
| Bargaining | $0.80 \pm 0.04$ | $0.18 \pm 0.30$ | $-0.50 \pm 0.64$ |
| Barista | $0.90 \pm 0.02$ | $0.45 \pm 0.18$ | $-0.79 \pm 0.83$ |
| Sharing Secrets | $0.78 \pm 0.05$ | $0.65 \pm 0.12$ | $0.55 \pm 0.19$ |
| Dating App | $0.71 \pm 0.06$ | $0.59 \pm 0.15$ | $0.37 \pm 0.27$ |
| White Lie During Date | $0.87 \pm 0.03$ | $0.49 \pm 0.18$ | $0.44 \pm 0.25$ |
| Marriage Argument | $0.82 \pm 0.04$ | $0.45 \pm 0.19$ | $0.62 \pm 0.18$ |
| **Mean** | $\mathbf{0.83 \pm 0.03}$ | $\mathbf{0.47 \pm 0.07}$ | $\mathbf{0.12 \pm 0.21}$ |

## B.2 MULTILEVEL MODEL

Using a multilevel model, we analyze the agreement between GPT-4 and human raters by assigning scenario as a random effect. We measure if candidates preferred in pairwise comparisons by the combined stakeholders had higher mean scores from human raters than those less preferred.

In the multilevel model, the independent variable was the weighted sum of stakeholder pairwise judgements, where each stakeholder evaluates each pair as better (1), worse (0), or tie (0.5) three times and takes the mean. Our dependent variable was the scores provided by human raters, which were on a 0–10 scale. The data has a three-level structure, with individual items nested within pairwise comparisons, which were further nested within the scenarios. The scenarios were treated as random effects to control for inherent differences among scenarios.

The modeling results revealed a significant fixed effect for the weighted pairwise judgements on the score provided by human raters (coef $= 0.427, p = 0.041$), providing strong evidence of significant agreement between the GPT-4 and human evaluations across scenarios. We found that this effect differed across scenarios, indicating the multilevel model's appropriateness in taking into account the hierarchical nature of our data.

## B.3 INTER-RATER RELIABILITY AND AGREEMENT ON UNORTHODOX ADVICE AND OTHER DATA SUBSETS

We compute the IRR for baseline generations, candidates conditioned on only normal advice, and candidates with at least one unorthodox advice (unorthodox candidates) and find that unorthodox candidates have a much lower IRR (see Table 7). First, this suggests that unorthodox advice is harder to simulate and judge consistently even across human agents. Additionally, this implies that EGS using any LLM trained on human-generated data would likely also be more inconsistent during simulated judgements.

With this insight, we analyze the agreement between both normal, one unorthodox, and both unorthodox candidate pairs using both the pairwise winner vs. loser and the percentage agreement metric (see Table 8). We find that comparisons between normal advice to have higher agreement than comparisons between unorthodox advice in both metrics, but note that the differences are small. Intriguingly, we find that comparisons between normal candidates and unorthodox candidates have a much higher agreement in both metrics, indicating a high agreement in both absolute difference in ratings (winner vs. loser) and proportion of ratings (percentage agreement).

## B.4 PERCENTAGE AGREEMENT METRIC

Agreement is calculated between $n$ mean human ratings and $\binom{n}{2}$ LLM pairwise comparisons. For each pair of candidates $i, j$, we calculate their mean human scores $h_i, h_j$, and break ties favoring the candidate with the higher mean score when normalizing within each participant. For each simulated

Table 8: Pairwise winner vs. loser and percentage agreement analysis find agreement is highest in comparisons between normal and unorthodox candidates, then normal vs. normal candidates, and lowest in unorthodox vs. unorthodox candidates. There were 153, 216, and 66 both normal, one unorthodox, and both unorthodox comparisons respectively per scenario.

| | Winner vs. Loser | | | Percentage Agreement | | |
|---|---|---|---|---|---|---|
| Compared Advice | Both Normal | One Unorthodox | Both Unorthodox | Both Normal | One Unorthodox | Both Unorthodox |
| Plane Crash | 0.38*** | 0.32*** | 0.28* | 0.63 | 0.63 | 0.62 |
| Product Launch | 0.11 | 0.55*** | 0.13 | 0.56 | 0.81 | 0.46 |
| Bargaining | 0.29*** | -0.36*** | -0.06 | 0.70 | 0.41 | 0.53 |
| Barista | -0.01 | 2.33*** | -0.04 | 0.54 | 0.75 | 0.53 |
| Sharing Secrets | 0.56*** | 0.88*** | 0.61*** | 0.73 | 0.82 | 0.79 |
| Dating App | -0.15 | -0.26*** | -0.15 | 0.46 | 0.37 | 0.44 |
| White Lie During Date | 0.01 | -0.15* | -0.27* | 0.47 | 0.41 | 0.37 |
| Marriage Argument | 0.13 | 0.45*** | 0.41** | 0.53 | 0.73 | 0.65 |
| **Mean** | **0.16** | **0.47** | **0.11** | **0.58** | **0.62** | **0.55** |

Table 9: Agreement between GPT-4 pairwise ratings and human scores (compared pairwise). We consider the set of all pairwise comparisons, as well as subsets where the absolute difference of the human ratings, $d$, differ by at least 0.5, 1, and 1.5 respectively. $n$ denotes the number of samples in the corresponding subsets.

| Scenario | Agreement | $n$ | $d \geq 0.5$ | $n$ | $d \geq 1$ | $n$ | $d \geq 1.5$ | $n$ |
|---|---|---|---|---|---|---|---|---|
| Plane Crash | 0.63 | 435 | 0.72 | 259 | 0.80 | 122 | 0.83 | 46 |
| Product Launch | 0.67 | 435 | 0.74 | 237 | 0.83 | 92 | 0.96 | 28 |
| Bargaining | 0.53 | 435 | 0.43 | 243 | 0.41 | 86 | 0.33 | 24 |
| Bargaining (-unorthodox advice) | 0.69 | 153 | 0.79 | 72 | 0.88 | 21 | 1.00 | 3 |
| Barista | 0.64 | 435 | 0.71 | 330 | 0.77 | 269 | 0.81 | 239 |
| Sharing Secrets | 0.78 | 435 | 0.84 | 312 | 0.91 | 194 | 0.94 | 100 |
| Dating App | 0.41 | 435 | 0.37 | 282 | 0.31 | 144 | 0.37 | 52 |
| White Lie During Date | 0.43 | 435 | 0.41 | 265 | 0.40 | 124 | 0.58 | 33 |
| Marriage Argument | 0.65 | 435 | 0.64 | 312 | 0.71 | 181 | 0.74 | 89 |

audience, we average over 3 generated comparisons between $i$ and $j$, and then compute a weighted sum across audiences using LLM-assigned weights. From this, we derive three classes - {majority prefers $i$, majority prefers $j$, balanced} - denoted $m_i$, $m_j$, and $m_b$ respectively.

$$\text{agreement} = \frac{1}{\binom{n}{2}} \sum_{i \neq j} A(i,j) \qquad A(i,j) = \begin{cases} 1 & \text{if } h_i > h_j, m_i \text{ or } h_i < h_j, m_j \\ 0.5 & \text{if } m_b \\ 0 & \text{otherwise} \end{cases} \qquad (2)$$

This metric can be interpreted as the expected percentage agreement between $\{h_i > h_j, h_i < h_j\}$ and $\{m_i, m_j\}$, with $m_b$ tiebreaked randomly. We do not adjust based on the amount of agreement expected by chance (such as in Cohen's $\kappa$) as the classes of our pairwise ratings are by definition unbiased since they only depend on the order in which the candidates are shown to the rater. In particular, we did not find evidence of any positional bias (Wang et al., 2023a), i.e., preferring Scenario 1 over Scenario 2 regardless of content, in the pairwise comparisons done by EGS (see Appendix B.11).

## B.5 AGREEMENT BECOMES MORE PRONOUNCED FOR EASIER CASES

We consider easier pairwise comparison cases where the difference in mean ratings between the two candidates are greater than certain thresholds, and find that agreement between humans and LLM-simulated audiences becomes more pronounced for both high and low agreement cases (see

Table 10: An ablation on the *Simulate* step shows that simulating audiences and choosing the best candidate performs better than selecting a candidate or set of advice at random for five scenarios and a subset of a sixth scenario. Standard deviations are shown for both candidates and advice sets.

| Scenario | Average candidate | EGS candidate | Average advice set | EGS advice set |
|---|---|---|---|---|
| Plane Crash | $6.03 \pm 0.66$ | **7.95** | $6.03 \pm 0.40$ | **6.37** |
| Product Launch | $6.03 \pm 0.57$ | **7.05** | $6.03 \pm 0.44$ | **6.68** |
| Bargaining | $\mathbf{5.95 \pm 0.54}$ | 5.85 | $\mathbf{5.95 \pm 0.43}$ | 5.63 |
| Bargaining (-unorthodox advice) | $6.20 \pm 0.45$ | **6.60** | $6.20 \pm 0.31$ | **6.57** |
| Barista | $4.08 \pm 1.84$ | **5.40** | $4.08 \pm 1.86$ | **5.55** |
| Sharing Secrets | $5.36 \pm 0.84$ | **5.55** | $5.36 \pm 0.78$ | **5.95** |
| Dating App | $\mathbf{5.41 \pm 0.68}$ | 5.05 | $\mathbf{5.41 \pm 0.57}$ | 4.98 |
| White Lie During Date | $\mathbf{6.75 \pm 0.61}$ | 6.70 | $\mathbf{6.75 \pm 0.39}$ | 6.37 |
| Marriage Argument | $6.18 \pm 0.79$ | **7.80** | $6.18 \pm 0.62$ | **6.88** |

Table 11: An ablation on the *Explore* step shows that searching the conceptual space for advice to condition our candidates upon creates higher-quality candidates than searching for unrelated or encouragement-type advice in six scenarios. Errors shown are standard errors of the mean. *, **, and *** denote $p < 0.05$, $p < 0.01$, and $p < 0.001$ when compared to the conceptual search space.

| Scenario | Conceptual Search Space | Irrelevant Search Space | Encouragement Search Space |
|---|---|---|---|
| Plane Crash | $\mathbf{6.03 \pm 0.07}$ | $1.92 \pm 0.11$*** | $3.91 \pm 0.14$*** |
| Product Launch | $\mathbf{6.03 \pm 0.09}$ | $5.82 \pm 0.10$ | $4.26 \pm 0.12$*** |
| Bargaining | $\mathbf{5.95 \pm 0.09}$ | $3.98 \pm 0.14$*** | $5.00 \pm 0.12$*** |
| Barista | $4.08 \pm 0.11$ | $2.85 \pm 0.14$*** | $\mathbf{5.84 \pm 0.12}$*** |
| Sharing Secrets | $\mathbf{5.36 \pm 0.09}$ | $4.91 \pm 0.11$** | $4.89 \pm 0.11$** |
| Dating App | $\mathbf{5.41 \pm 0.08}$ | $4.65 \pm 0.11$*** | $5.34 \pm 0.11$ |
| White Lie During Date | $\mathbf{6.75 \pm 0.08}$ | $3.62 \pm 0.13$*** | $5.57 \pm 0.13$*** |
| Marriage Argument | $5.41 \pm 0.10$ | $4.39 \pm 0.14$*** | $\mathbf{6.45 \pm 0.13}$ |

Table 9). We calculate agreement using the percentage agreement metric proposed in Section 5.3. For scenarios starting with agreement $> 0.6$ (plane crash, product launch, bargaining (-unorthodox), barista, sharing secrets, marriage argument), as we increase the threshold agreement consistently increases as well. In particular, five of these reach $0.8$ agreement with moderate sample sizes. For the three scenarios that begin with agreement $<0.6$, agreement generally continues to decrease as the threshold increases, indicating a fundamental misalignment between human and simulated audience judgments.

### B.6 SIMULATED AUDIENCE EVALUATIONS SELECT HIGH-PERFORMING CANDIDATES

We perform an ablation on the effectiveness of the *Simulate* step by comparing the best candidate and advice of EGS against a baseline that selects a random candidate from the *Generate* step. In five scenarios and the bargaining scenario without unorthodox advice, EGS selects a more optimal candidate and set of advice than the baseline (see Table 10).

We note that the three scenarios that the *Simulate* step performs poorly at are also the scenarios which had low human-GPT agreement, which matches the intuition that the accuracy of the comparison process heavily determines the performance of the chosen outcome.

### B.7 ABLATION: WHICH SEARCH SPACE TO EXPLORE?

We compare our search over conceptual advice with encouragement-type advice and irrelevant advice across all scenarios, and find that conceptual advice generates better candidates on average for six of the eight scenarios (see Table 11). To construct the encouragement and irrelevant versions, we

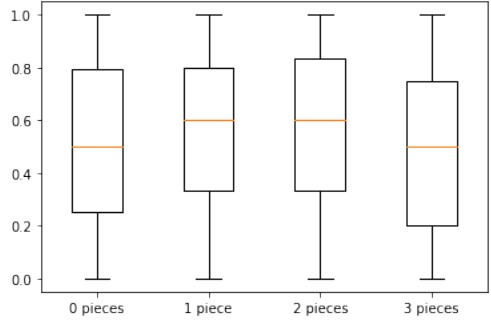

Figure 2: Average human scores of pilot candidates conditioned on different amount of advice. Each participant rated 20 candidates, and scores are normalized within each participant.

modify the *Explore* step such that GPT-4 is asked to generate advice that encourages the communicator instead of providing strategic value and advice irrelevant to the scenario, respectively.

The mean score of candidates in the conceptual search space is greater than the encouragement and irrelevant search spaces in 6/8 and 8/8 of the scenarios respectively, with 5 and 7 scenarios being highly statistically significant ($\alpha < 0.01$). However, conceptual advice also performs significantly worse than encouragement advice in the barista scenario, suggesting that the optimal advice prior might vary based on the specific setting.

### B.8 HOW EFFECTIVE IS UNORTHODOX ADVICE?

We compare the human scores of candidates that utilize unorthodox advice with those that do not, and find that those that contain unorthodox advice perform better in 4 of the 8 scenarios, with 2 scenarios having statistically significant differences ($\alpha = 0.05$). We also find that 2 of the scenarios' highest human-rated candidates are conditioned on the unorthodox advice, showing that it has the potential to create better communication options as well. In particular, we note that since the maximum amount of advice per candidate is limited to 2, the unorthodox advice is actively replacing a normal piece of advice when it does better.

### B.9 PILOT RESULT: PERFORMANCE DECREASES WITH TOO MANY PIECES OF ADVICE

One potential benefit for large language models is that it is able to consider a much larger set of advice than humans can. With this in mind, we conducted an initial study with one scenario where candidates could be based on up to three pieces of advice. We found a large drop in performance when three pieces of advice were incorporated: 0.47 vs. 0.55 and 0.56 for 1 piece and 2 pieces respectively[2] when three pieces of unique advice were incorporated (see Figure 2). Qualitatively, we observed a common behavior where GPT-4 attempts to incorporate all of the advice into the generated message body, resulting in utterances that were conceptually disconnected and excessively long.

Based on this result, we limited the maximum number of advice for any candidate in the *Generate* step to two in our main experiments. This is also intuitively aligned with the applications of the model. In realistic improvisation scenarios, a human agent might find it much easier to effectively use less pieces of advice (Miller, 1956). However, as the capabilities of LLMs continue to improve, we may see changes in the behavior of these models that may allow for the utilization of many more pieces of advice at once.

### B.10 ABLATION: MULTIPLE ADVICE IMPROVES HIGHEST-SCORING CANDIDATES

We investigate whether having multiple pieces of advice per candidate is beneficial for performance, and find that incorporating multiple pieces of advice per candidate 1) does not generally improve

---

[2]These scores were normalized across each reviewer, and are thus in $[0, 1]$.

Table 12: Having multiple advice per candidate does not consistently improve the mean candidate score, but often contributes to improving the best candidate. Errors shown with mean scores are standard errors of the mean, and *, **, *** denote significance at $p = 0.05, 0.01,$ and $0.001$ respectively.

| Scenario | Mean Score Single Advice | Mean Score Multiple Advice | Highest Score Single Advice | Highest Score Multiple Advice |
|---|---|---|---|---|
| Plane Crash | **6.25 ± 0.20*** | 5.88 ± 0.14 | **7.95** | 7.05 |
| Product Launch | 5.71 ± 0.15 | **6.24 ± 0.12**** | 6.55 | **7.05** |
| Bargaining | 5.77 ± 0.11 | **6.06 ± 0.14** | 6.30 | **7.20** |
| Barista | **4.83 ± 0.53***** | 3.57 ± 0.40 | 6.20 | **6.25** |
| Sharing Secrets | **5.38 ± 0.31** | 5.34 ± 0.16 | **7.30** | 6.60 |
| Dating App | **5.55 ± 0.17*** | 5.20 ± 0.17 | 6.40 | **6.50** |
| White Lie During Date | **6.80 ± 0.20** | 6.72 ± 0.13 | 7.95 | **8.00** |
| Marriage Argument | **6.32 ± 0.27** | 6.09 ± 0.16 | **7.80** | 7.45 |

the mean performance, but 2) does improve the performance of the highest-performing candidate. In Table 12, we provide the average scores and highest scores for candidates generated using 1 and 2 pieces of advice. We observe that, even though single advice performs better on average than multiple advice (but not generally significantly so), multiple advice is responsible for the human-rated top candidate in five of the eight scenarios. This demonstrates the benefit of conditioning candidates on combinatorial sets of advice in the *Generate* step.

### B.11 SIMULATED AUDIENCE JUDGEMENTS CONTAIN NO EVIDENCE OF POSITIONAL BIAS

In past work (Wang et al., 2023a), LLMs are shown to have a tendency of preferring some options over others simply due to the order they appear in the list of options. We conduct an analysis on whether this behavior can be detected in pairwise comparisons done by simulated audiences. Across the eight scenarios, we assign "prefer scenario 1" with value 1, "prefer scenario 2" with value 0, and "tie" with value 0.5, similar to Equation 1. We take the average over all pairwise comparisons and find a mean of $0.5011$ and a standard error of the mean of $0.0037$. This leads us to conclude that there is no evidence of a positional bias in the pairwise comparisons of simulated audiences.

### B.12 GENERATING CANDIDATES CONDITIONED ON STAKEHOLDERS

In this section, we investigate the efficacy of generating candidates not only based on advice, but also catered to specific stakeholder audience profiles. We first cover the methods and prompts we use, followed by the generated candidates and their comparison results against the best candidate in the normal EGS framework.

#### B.12.1 METHODS AND PROMPTS

We evaluate on the plane crash scenario, where there are two stakeholder audiences, General Public and Victims' Families. For each stakeholder, we modify the original message candidate generation prompt by inserting elements of the stakeholder's description and the question they are asked to decide between the pairwise comparison.

After the candidates are generated, we evaluate whether these new candidates are more preferred by the same simulated audiences using GPT-4. We use the exact same comparison prompt as before (listed in Appendix A.5.2) for the pairwise comparisons, which includes elements that we incorporated into the new candidate generation prompt.

**System Prompt:** (stays the same)

You are an airline spokesperson. You want to minimize the negative public opinion of your company.

**User Prompt (original):**

Earlier today, a flight of the airline company you represent crashed due to unknown causes and there were 2 passengers that died. Right now, you are about to make the first public statement to the media at a live press conference. You only have 30 seconds of airtime to speak about the issue. You remember a piece of advice, "Express sincere empathy and condolences towards the victims and their families to show that your company holds human life in the highest regard.", and decide to focus on using it during your statement. What do you say? Answer:

**User Prompt (conditioned on General Public):**

Earlier today, a flight of the airline company you represent crashed due to unknown causes and there were 2 passengers that died. Right now, you are about to make the first public statement to the media at a live press conference. You only have 30 seconds of airtime to speak about the issue. You remember a piece of advice, "Express sincere empathy and condolences towards the victims and their families to show that your company holds human life in the highest regard.", and decide to focus on using it during your statement.

You are aware of the reactions that everyday citizens watching the live press conference on TV might have depending on what you say, especially on whether they would maintain a favorable opinion of the airline and consider using the airline's services in the future.

What do you say? Answer:

**General Public System Prompt:** You are an everyday citizen watching a live press conference on TV regarding a recent airline crash [...]

**General Public User Prompt:**

Scenario 1: [candidate 1]

Scenario 2: [candidate 2]

In which scenario would you be more likely to maintain a favorable opinion of the airline and consider using their services in the future? [...]

**User Prompt (conditioned on Victims' Families):**

Earlier today, a flight of the airline company you represent crashed due to unknown causes and there were 2 passengers that died. Right now, you are about to make the first public statement to the media at a live press conference. You only have 30 seconds of airtime to speak about the issue. You remember a piece of advice, "Express sincere empathy and condolences towards the victims and their families to show that your company holds human life in the highest regard.", and decide to focus on using it during your statement.

You are aware of the reactions that victims' families watching the live press conference might have depending on what you say, especially on whether they feel empathized with and reassured about the airline's commitment to rectify the situation.

What do you say? Answer:

**Victims' Families System Prompt:** You are a family member of one of the victims in the recent airline crash. You're watching a live press conference, [...]

**Victims' Families User Prompt:**

Scenario 1: [candidate 1]

Scenario 2: [candidate 2]

In which scenario would you be more likely to feel empathized with and reassured about the airline's commitment to rectify the situation? [...]

### B.12.2 GENERATED CANDIDATES AND COMPARISON RESULTS

We find that none of the new candidates are preferred by either audience over the best candidate in any of the 36 comparisons. Below, we provide the texts for the original best candidate and each of the six new candidates generated based on either the General Public or Victims' Families stakeholders

(3 each). For each new candidate, we provide three simulated preferences from each audience on which option they prefer.

We note that our results are limited to GPT-4 comparisons, and that we might see different results in human evaluations. However, it remains that if conditioning on stakeholders were added to EGS, the framework would select the original best candidate to recommend instead of any generated in the new method. We also note that our result does not imply that generating candidates based on stakeholders is bad or inferior on average; it only demonstrates that the best candidate of the original EGS framework is more promising than this alternative in the scenario.

**Original best candidate, no stakeholder audiences considered**

Ladies and Gentlemen, we are absolutely devastated about the tragic incident that occurred today. We extend our heartfelt condolences and deepest sympathies to the families who lost their loved ones in this unfortunate event. The value of human life is paramount, and our thoughts are with all those affected by this incident. We're committed to investigating the event thoroughly, cooperating fully with aviation authorities to ensure this does not happen again. As information becomes available, we will promptly and transparently share updates. Our priority now is supporting the affected families during this incredibly difficult time. Thank you.

**Conditioned on: General Public (response #1)**

First and foremost, our hearts truly go out to the two individuals and their families affected by today's incident. We cannot begin to imagine the level of grief they are experiencing, and we would like to extend our deepest sympathies. Your safety and well-being is our topmost priority and we are fully committed to finding out what happened and making sure that it doesn't happen again. We will leave no stone unturned in this investigation, working closely with the relevant authorities. Your trust and safety are paramount to us. Thank you.

**Comparison results against best candidate:**

General Public: loss, loss, loss

Victims' Families: loss, loss, loss

**Conditioned on: General Public (response #2)**

On behalf of our airline, we extend our deepest condolences to the families affected by this unbelievable tragedy. We are profoundly saddened by this incident. The safety and concern of our passengers is paramount. Our priority now is supporting the investigations to determine the cause, as we owe this to the families and loved ones of those we lost today. We will do everything in our power to ensure that this incident is thoroughly investigated, and intend to learn from it how to preclude such a situation in the future for the safety and trust of our valued passengers. Let's stand together in this difficult time.

**Comparison results against best candidate:**

General Public: loss, loss, loss

Victims' Families: loss, loss, loss

**Conditioned on: General Public (response #3)**

Ladies and Gentlemen, we are deeply saddened by today's unfortunate incident. Our hearts, thoughts, and deepest condolences extend to the families and loved ones of the two passengers we've tragically lost. We want to assure you that we place the utmost importance on the safety of our passengers and we are working tirelessly with authorities to understand the causes. We deeply appreciate your understanding and patience during this challenging time. Our commitment to safety and service remains unwavering. We appreciate your continued trust and support.

**Comparison results against best candidate:**

General Public: loss, loss, loss

Victims' Families: loss, loss, loss

**Conditioned on: Victims' Families (response #1)**

Our deepest sympathies go out to those affected by today's tragic incident. We are devastated by the loss of our passengers and fully committed to providing support for their loved ones during this difficult time. As an airline, the safety and care of our passengers are paramount and we are working tirelessly to determine the cause of the incident. We assure you that we will learn from this and take measures to prevent such an accident from happening again. Our thoughts are with the families and friends of the victims, and we extend our most heartfelt condolences.

**Comparison results against best candidate:**

General Public: loss, loss, loss

Victims' Families: loss, loss, loss

**Conditioned on: Victims' Families (response #2)**

On behalf of our company, I extend our deepest sympathies to the families of those impacted by this tragic event. We're immensely heartbroken and we share your pain in this difficult time. The safety of our passengers has always been, and will always be our utmost priority. We are working diligently with authorities to fully investigate the cause of the unfortunate occurrence to ensure this doesn't happen again. Please be assured, we are committed to you and will do everything within our capacity to support and assist those affected.

**Comparison results against best candidate:**

General Public: loss, loss, loss

Victims' Families: loss, loss, loss

**Conditioned on: Victims' Families (response #3)**

We are deeply sadden by the tragic incident that occurred earlier today. Our heartfelt condolences go out to the families and loved ones of the passengers involved. The loss of any life is devastating and we utterly understand the trauma and grief the families are going through. At this moment, our utmost priority is to support the affected families and cooperate with authorities to immediately determine what led to this incident. We assure you that we hold passenger safety and human life as our highest responsibility. Please join us in praying for the departed souls and their grieving families.

**Comparison results against best candidate:**

General Public: loss, loss, loss

Victims' Families: loss, loss, loss

### B.13 AGGREGATION OF STAKEHOLDER OPINIONS

In this section, we conduct a preliminary investigation into alternative aggregation methods of stakeholder opinions. Our scenarios can include multiple LLM-generated stakeholders, and weights assigned to each by the LLM. However, a weighted sum is not necessarily the best approach for aggregating stakeholder opinions together. For instance, if all stakeholders are able to severely affect the communicator's goal, then they might opt for a strategy to maximize the lowest score across all stakeholders. Alternatively, if you only need one stakeholder's approval to achieve your goal, a viable strategy might be to maximize the highest score across all stakeholders.

In the cognitive science literature, Gates et al. (2020) show that when humans are trying to be helpful to multiple people at once, their behavior is best described by the *maximin* metric, describing the desire to maximize the happiness of the worst-off person, while also consistent with maximizing group utility (*maxsum*) and equality (*inequality aversion*) to a lesser extent.

In our three scenarios with multiple stakeholders – product launch, plane crash, and dating app, we find that these metrics agree with each other in all but two cases on the best advice and one case on the best candidate. Notably, the *maximin* metric performs worse on the plane crash scenario, selecting the 17th best candidate (out of 30) instead of the 1st, and the 3rd best advice set instead of the 2nd. In the future, more investigations can be done to determine which of the aggregation metrics make the most sense on top of LLM comparisons.

### B.14 COMBINING CANDIDATES AFTER SIMULATED COMPARISONS

One potential alternative to the current framework is to combine together top-performing candidates in the *Simulate* step to potentially create an even better recommendation. We conduct a preliminary investigation on this method.

Specifically, we take the top $k$ candidates from the simulated comparisons, as well as the $\binom{k}{2}$ comparisons between them, and input them all into the LLM. Then, we ask the LLM to construct a new candidate message that performs better based on these comparisons. Note that in practice, $k$ should be small to avoid scalability issues - in our experiments we use $k = 2$, but the code available allows for any $k$ to be selected.

**Prompt.**

[Scenario description]

Here are [# candidates] possible candidates for [communicator] to use:
[list of top-performing candidates]

Pairwise comparisons from [audience]:

(repeat for $\binom{k}{2}$ pairs of candidates)

    Question asked to [audience]:

    [prompt used for stakeholder audience]

    We have three comparison results from [audience]:

    [reasoning + answer 1 from simulated audience]

    [reasoning + answer 2 from simulated audience]

    [reasoning + answer 3 from simulated audience]

Based on the pairwise comparisons between the candidates, construct a new candidate message that performs better based on these comparisons.

New candidate:

**Generated candidates.** We observe that the new candidates try to incorporate information across both candidates, and are thus much longer than their original counterparts. However, this often makes their utterances too lengthy for the scenario.

For instance, in the white lie during date scenario, the two original candidates are:

1. "Wow, I must say that style isn't easy to pull off, but you certainly can! You just light up in whatever you wear and that joy is contagious. It's not everyone who can wear a jacket like that and make it work. You make it feel like more than just a jacket, you bring it to life!"

2. "Wow, I love how that jacket brings out the sparkle in your eyes, and I can see how happy it makes you feel. It certainly adds a unique flair to your whole look."

And the new candidate:

"Wow, that jacket sure is unique! And trust me, it's not everyone who can wear something like that and bring it to life. The way it brings out the sparkle in your eyes is really captivating. Above everything, what I love the most is seeing how happy it makes you feel. Your joy really is

contagious, and it lights up your whole look!"

Similarly, in the barista scenario, the two original candidates are:

1. "Sure, a medium whole milk latte coming up! That's one of my favorite drinks too, it's really smooth and flavorful. And good choice with the whole milk - it gives the latte such a creamy texture, plus it's our freshest batch. I'm sure you will enjoy it!"
2. "Absolutely, I'd be pleased to prepare a medium latte for you with our freshest whole milk. It lends the latte a very creamy texture. Interestingly, that's also one of my favorite drinks. Its smoothness and rich flavor are truly satisfying. I'm sure you're going to love it!"

Meanwhile, the new candidate:

"Sure thing! I'll get started on your medium latte with whole milk right away. That's actually one of my personal favorites as well - it's incredibly smooth with a delightful rich flavor. Choosing the whole milk really makes all the difference, doesn't it? It creates such a wonderful, creamy texture, and you're in for a treat since we actually just received our freshest batch. I have a feeling you are going to absolutely love this drink!"

Though the construction of the new candidates do seem influenced by the pairwise comparisons, speculatively, they seem too long for the scenario and thus feel inadequate to use in practice. Future investigations can be done to see if these outcomes are indeed less preferred by human raters, while also exploring how LLMs can better combine message candidates in a concise manner.

### B.15    HALLUCINATION ANALYSIS

We conduct a preliminary qualitative hallucination analysis to investigate the types of hallucinations EGS might make in the eight scenarios.

**Plane Crash.** In this scenario, we observe 2 of the 30 candidates have hallucinations, both being influenced by the advice, "Reassure the public that this is an isolated event and emphasize on your airline's long-standing reputation for safety and commitment to continuous improvement". The two candidates exaggerate this by stating that "We have always been lauded for our commitment to safety" and "We want to reassure everyone that this is an unprecedented and isolated event", which are both not guaranteed to be true. However, the degree of these hallucinations are minor.

**Product Launch.** In this scenario, we observe 0 hallucinations in the 30 candidates. The only caveat is that this is under the assumption that the advice "declare a limited time 24-hour sale where the first 100 customers get the phone at an additional 50% off" is followed in those scenarios.

**Bargaining.** In this scenario, we observe 8 candidates with hallucinations out of 30 total. All eight are caused by the unorthodox piece of advice, "Jill, consider offering to promote the artisan's work on your social media in exchange for a discount on the vase", where the LLM infers that Jill has a 'considerable', or 'a good number' of followers on social media when this isn't actually guaranteed to be true.

**Barista.** In this scenario, the unorthodox advice, "did you know customers say our lattes taste better if you pay in tips?" can be interpreted as a friendly joke, and therefore its use is generally not considered a hallucination. However, in one case, the LLM adds that this is "a fun little tradition we have here", which moves towards a claim rather than friendly banter. Thus, we have 1 hallucination out of 30 candidates.

**Sharing Secrets.** In this scenario, we observe 3 candidates with hallucinations out of 30 total. Two of these take the known statement, "You used to be impulsive when spending money and you still have some debt" and hallucinate that this occurred in their early twenties. The last one was also about spending money, hallucinating that "it's not a shopping spree at a fancy boutique I'm talking about, it's more like a Pokémon cards obsession".

**Dating App.** We observe that 21 candidates of 30 contain hallucinations, including personality traits (e.g., liking dad jokes), hobbies (e.g., reading), trivia (e.g., his dog liking caramel ice cream), and stories about his dog. The stories are due to the advice, "Share a fun, unique story involving Scott,

your beagle...", which necessitates for hallucination and appears in 9 candidates. The remaining hallucinations can be remedied by the inclusion of more details about the user in the input to EGS, but the reduction of authenticity in the user's profile when these traits are underspecified is something users must be wary of.

**White Lie During Date.** In this scenario, we observe 0 instances of hallucinations across the 30 candidates. All utterances are reasonable compliments that do not assume any unnecessary factual information.

**Marriage Argument.** In this scenario, we found 2 instances of hallucinations across the 30 candidates. Both were attributable to the unorthodox advice, "Start your conversation by asking about his day and his friends' shenanigans, showing genuine interest...", with the utterances being "Did Pete manage to pull off that ridiculous trick he was talking about the other day?" and "Did Mike finally ace his pool game? How was James handling his break up?".

**Summary.** Across the scenarios, we observe that many of the hallucinations do not detract from the overall idea of the candidates. Thus, when EGS is used in practice, these hallucinations would just be replaced by the user and serve as benign templates. For example, "Did Mike finally ace his pool game?" can be replaced by anything interesting that the user remembers about their husband's friends.

The only exceptions to this are when the advice themselves make assumptions that do not align with reality. For example, in the plane crash scenario, if the airline company does not have a long-standing reputation for safety, the corresponding candidates generated may be unusable. The same is true if the user does not have a social media following in the bargaining scenario. However, these hallucinations are rare, and incorporating more information into the scenario description is a quick fix in the case that unusable advice are generated.

## C  SHP EXPERIMENTS

Table 13: Prompt used in the CoT setting.

User:
Comment 1:
<Comment 1>
Comment 2:
<Comment 2>
Post:
<Post>
Given the post, choose the comment that you are more likely to upvote.
Please think step by step and explain your reasoning. Stop after you output the final answer.
Put your response in the following format:
Reason: [reasoning text]
Answer: Comment [1 or 2]
End

### C.1  EXPERIMENT DETAILS

Following the steps taken by the original authors of the SHP dataset[3], we first filter the dataset such that all pairwise comments have a ratio of at least 3 in the number of upvotes. This aims to reduce the amount of noise in the dataset and ensure that one comment is strongly preferred over the other. Furthermore, we randomly assign the order of the comments such that the chance of any comment being either the first or the second comment is 50%. This reduces possible positional biases from the model. For generation, we use hyperparameters `top_p=0.9` and `temperature=0.1`.

---

[3] `https://huggingface.co/datasets/stanfordnlp/SHP`

Table 14: Prompt used in the Redditor Simulation – Default setting for the askculinary domain.

---

System:
You are interested in all culinary-related things.
You are currently browsing a reddit culinary forum, and you are looking for interesting content to read.
You click on a post and you are reading through the comments.

---

User:
Comment 1:
<Comment 1>
Comment 2:
<Comment 2>
Post:
<Post>
Given the post, choose the comment that you are more likely to upvote.
Please think step by step and explain your reasoning. Stop after you output the final answer.
Put your response in the following format:
Reason: [reasoning text]
Answer: Comment [1 or 2]
End

---

Table 15: Prompt used in the Redditor Simulation – Funny setting for the askculinary domain.

---

System:
You are interested in all culinary-related things.
You are currently browsing a reddit culinary forum, and you are looking for interesting and entertaining content to read.
You are more likely to upvote comments that are funny and that can make you laugh.
You click on a post and you are reading through the comments.

---

User:
Comment 1:
<Comment 1>
Comment 2:
<Comment 2>
Post:
<Post>
Given the post, choose the comment that you are more likely to upvote.
Please think step by step and explain your reasoning. Stop after you output the final answer.
Put your response in the following format:
Reason: [reasoning text]
Answer: Comment [1 or 2]
End

---

To better interpret our results, in asksocialscience[4], the forum's rules include "all claims in top level comments must be supported by citations to relevant social science sources" and "comments must be serious attempts to answer the question".

## C.2 Effect of Redditor profiles

To further test the effect of different prompts on the ability of the LLM to simulate Redditors, we focus on the askculinary domain and test four additional profile settings. Each of the four additional profiles is a variation of the "funny" profile, where we perturb individual sentences to obtain different profiles. These additional profiles can be found in Table 18.

We notice that the performance of the LLM's ability to simulate Redditors increases for each of the variations of the profile, as shown in Table 19. In particular, variation 1 and 4 saw a relatively large increase while the two other remained mostly the same. Specifically, when we change the first

---

[4] https://www.reddit.com/r/AskSocialScience/

Table 16: Example of the output from the three different settings in the askculinary domain.

**Post**:
When setting up a breading station, is it better to season the flour or panko? For example when you're breading proteins such as chicken or pork, do you season the flour or the panko? I've always just seasoned the flour but was curious to hear what other people do.

**Comment 1 (42 upvotes)**:
100% read this as "breeding station". Looked at subs name. Asked myself what the fuck was wrong with people. Reread the title. Whats wrong with people is that some of us can't read. As to your question. I do both, though they are usually lightly seasoned.
**Comment 2 (9 upvotes)**:
I season all steps.

**Chain-of-Thought** Output

Reason: Comment 2 is more relevant and directly answers the question posed in the post. It provides a clear opinion on the topic of seasoning in a breading station. Comment 1, while humorous, doesn't provide a clear stance on the question and spends more time discussing a reading error.
Answer: Comment 2

**Redditor Simulation – Default** Output

Reason: Comment 1 provides a humorous anecdote before answering the question, which can be entertaining but doesn't directly address the post's question until the end. Comment 2, on the other hand, is straightforward and directly answers the question, providing a clear stance on the topic. As someone interested in culinary-related things, I would appreciate the direct and relevant response more.
Answer: Comment 2

**Redditor Simulation – Funny** Output

Reason: Comment 1 is more likely to be upvoted because it contains a humorous anecdote about misreading the post title, which adds an entertaining element to the discussion. While both comments provide an answer to the question, Comment 1 does so in a way that is more engaging and likely to elicit a laugh.
Answer: Comment 1

sentence from "You are interested in all culinary-related things." to "You are interested in learning more about niche cuisine and cooking methods.", we see a large increase in performance, which suggests that users of the askculinary domain are more likely to be interested in "niche cuisine and cooking methods" as opposed to general culinary topics.

# D  EXTENDED RELATED WORK

**Prompt engineering.** Various prompting methods have been developed to improve the performance of LLMs. These include Chain-of-Thought (Wei et al., 2022), thinking step-by-step (Kojima et al., 2023), and self-consistency (Wang et al., 2023b). Jain et al. (2023) extends self-consistency to open-ended generation tasks such as code generation. While these works focus on editing prompts to better perform on NLP tasks and benchmarks, our paper focuses on the application of improving interpersonal communication at large.

**Human preferences.** Recent LLMs are trained to following instructions with reinforcement learning using human feedback (RLHF) (Ouyang et al., 2022). To train the reward model in RLHF, they collect human preference data from a team of human annotators (Ouyang et al., 2022; Touvron et al., 2023; Bai et al., 2022). While previous work use human preference data to train reward models that enable RLHF, we leverage them to test the ability of our model to adapt to scenarios that internet users have encountered (Ethayarajh et al., 2022).

Table 17: Example of the output from the three different settings in the asksocialscience domain.

---

**Post**:
Why is the US so dominating in entertainment and culture? I'm from Norway, not the largest country in the world, and I was thinking about where my entertainment comes from. Speaking a small language, I'm accustomed to speaking or listening to another language when I interact online or watch/listen to other entertainment mediums, and I'd guess that most non-english speaking people would do the same. But it seems like most of my entertainment comes from the US, something that I find weird when I think about the size of the country. Europe has more than double the population, and by that logic why isn't 2/3 of my entertainment from the continent? In global terms, the US is roughly 5% of the world population, yet it's dramatically overrepresented in most media...

**Comment 1 (5 upvotes)**:
I'm from Russia and puzzled about it as well. Speaking about TV, for example Ž014 I never watch television here, because it's 99% shit, but regurarly download latest episodes of Breaking Bad, Homeland and other great american shows... What I don't understand, personally, is how they ended up making stuff that's so much better than what we have here.

**Comment 2 (16 upvotes)**:
First: The Division of Labor is limited by the extent of the market. English language extends to a much larger number of wealthy people than any other language, which means there can be more differentiation within the English language market than other language markets... Second: Agglomeration effects and economies of scale*. Certain types of art thrive by being done at a place where other types of that art are being done. Obviously, if I want to make a movie the cheapest place to produce it will be Hollywood (filming it is a different question), because there are people and equipment there who specialize in making movies... Economies of Scale indicate the range of production where average cost is decreasing...

---

**Chain-of-Thought** Output

---

Reason: Comment 2 provides a detailed and well-thought-out explanation to the question posed in the post. It uses economic principles to explain why the US dominates in entertainment and culture, and gives examples to illustrate these principles. It also addresses the question from different angles, including the influence of language and the role of specific locations in the production of art. On the other hand, Comment 1 is more of a personal opinion and doesn't provide a comprehensive answer to the question.
Answer: Comment 2

---

**Redditor Simulation – Default** Output

---

Reason: Comment 2 provides a detailed and well-thought-out response to the original post. It offers a comprehensive explanation of why the US dominates in entertainment and culture, using concepts such as the division of labor, agglomeration effects, and economies of scale. It also provides examples to illustrate these concepts, making the explanation more understandable. On the other hand, Comment 1 is more of a personal anecdote and doesn't really answer the question posed by the original post.
Answer: Comment 2

---

**Redditor Simulation – Funny** Output

---

Reason: While both comments provide insightful perspectives on the question posed by the post, Comment 2 offers a more detailed and comprehensive explanation. However, considering my preference for humor and entertainment, Comment 1 is more likely to get my upvote. It is more relatable and has a touch of humor in the way the commenter compares the quality of TV shows in their home country to American shows.
Answer: Comment 1

---

# E EXTENDED DISCUSSION

## E.1 EXTENDING THE FRAMEWORK WITH USER CONTROLS

While we focus on having the LLM generate potential audiences, as mentioned above, there may be situations where an end-user might want to control the audience members and their weights more explicitly. For example, if an airline company did want to put together a statement about a crash, and knew that their primary audience was "family members" and a "regulatory board" that determines what statements are not permitted, they can insert one or more custom designed audience members into the framework, or edit an existing audience profile to their liking. Similar kinds of user controls

Table 18: Prompt used in the Redditor Simulation – Variations of the Funny setting for the askculinary domain.

| **Original Funny Setting** |
|---|
| System:
You are interested in all culinary-related things.
You are currently browsing a reddit culinary forum, and you are looking for interesting and entertaining content to read.
You are more likely to upvote comments that are funny and that can make you laugh.
You click on a post and you are reading through the comments. |
| **Funny Setting Variation 1** |
| You are interested in learning more about niche cuisine and cooking methods.
You are currently browsing a reddit culinary forum, and you are looking for interesting and entertaining content to read.
You are more likely to upvote comments that are funny and that can make you laugh.
You click on a post and you are reading through the comments. |
| **Funny Setting Variation 2** |
| You are interested in all culinary-related things.
You are currently browsing a reddit culinary forum, and you are looking for content to read.
You are more likely to upvote comments that are funny and that can make you laugh.
You click on a post and you are reading through the comments. |
| **Funny Setting Variation 3** |
| You are interested in all culinary-related things.
You are currently browsing a reddit culinary forum, and you are looking for interesting and entertaining content to read.
You are more likely to upvote comments that are creative or unexpected.
You click on a post and you are reading through the comments. |
| **Funny Setting Variation 4** |
| You are interested in all culinary-related things.
You are currently browsing a reddit culinary forum, and you are looking for interesting and entertaining content to read.
You are more likely to upvote comments that are funny and that can make you laugh.
You click on a post, read it and find it interesting, and start looking through a few comments. |

Table 19: Results of using variations of the Funny prompt

|  | askculinary |
|---|---|
| CoT | 69.0 |
| Default | 60.0 |
| Funny | 70.5 |
| Variation 1 | 77.0 |
| Variation 2 | 71.0 |
| Variation 3 | 71.0 |
| Variation 4 | 76.0 |

can limit or change the pieces of advice used within EGS, the types of candidates generated, and even the outcomes of the simulated comparisons.

Taken further, this also enables a human-in-the-loop version of EGS where any of the steps can be fully reviewed and/or amended by a human viewer. We expect this option to be structured as a trade-off between performance and ease of use, where a paradigm that incorporates more human intervention into EGS may see an increase in quality and alignment at the cost of increased human effort.

## E.2 EXTENDING TO MULTI-TURN CONVERSATIONS

While EGS primarily involves the optimization of a single message, it can be extended to simulate multi-turn dialogues of arbitrary length by simply extending the *Generate* step to include one or more responses from the audience and repeating for the duration of the dialogue (visually, the acronym becomes EGG…GS). At the end, we can *Simulate* the audiences' reactions to the entire

chain of generated dialogue and compare them to find the best candidate at each utterance opportunity and the best overall advice.

One can also envision a variant where the search space is re-explored at every single *Generate* step (visually, EGEG...EGS). This allows the advice to change based on the flow of the conversation, which could potentially lead to better performance but may not be realistic for live communication scenarios.

A limitation in the above approaches is that they generate an exponential amount of potential candidates with respect to the length of the dialogue, resulting in scaling issues. We can potentially amend this by inserting a *Simulate* step after every certain number of *Generate* steps, where each *Simulate* step selects only the top $k$ candidates to continue generating upon, essentially implementing beam search on top of the dialogue trajectories.

### E.3 APPLICATION: REASONING ABOUT THE PAST

Aside of optimizing communication in the present or preparing advice for the future, EGS can also be applied to reason about past communication events. In particular, we are able to perform counterfactual reasoning (CFR): Given a past scenario and its outcome, CFR concerns whether an alteration to the antecedent of the counterfactual affects the outcome (Pearl, 2000). While Ma et al. (2023) also use LLMs to do counterfactual reasoning, they focus on improving moral reasoning, whereas we propose the use counterfactuals to reason about causal effects in communication settings.

Specifically, given a past communication setting, we can use the *Explore* and *Generate* components to create a diverse list of alternatives to the antecedent, and then *Simulate* the outcomes when we replace the antecedent with each alternative. In this particular application, we can further improve the *Simulate* step by including the communication utterance used and the actual outcome as a gold standard example in the context provided to the LLM. Then, using the simulated pairwise comparison results, we can make conclusions in the simulated space about which utterances the communication could have used to reach a better outcome, or if any underlying pieces of advice were responsible for a particular type of outcome. Though these causal effects may not be directly transferrable to the real world due to simulation inaccuracies, they provide testable hypotheses that can be directly implemented into real subject experiments.

### E.4 APPLICATION: HUMAN STUDIES AND RLHF

Aside of suggesting a communication utterance or advice, EGS can also be used directly for its simulated human preferences or feedback. In particular, we highlight two potential use cases in human subject studies and RLHF.

By viewing the instructions of human subject studies as "communication" with the goal of collecting high-quality data, EGS can be used to design and test a manifold of human user studies. Specifically, we can *Explore* and *Generate* different study design protocols, and then *Simulate* participants to collect study data. Importantly, the simulated reactions of participants can be decoupled into (1) data collected for the study and (2) preferences on which study design is better, allowing for both design optimization and data collection at once. Many existing works show that LLMs are effective proxies of humans in research studies (see Section 2). However, these works stop at using LLMs to generate data/feedback, whereas EGS is also able to test and automatically select a best set of participant instructions as well.

Unlike RLHF, which requires training explicit reward models (Ouyang et al., 2022), EGS can generate audience members that function in some scenarios like reward models that rate generations. Furthermore, EGS is able to simulate audiences conditioned on different backgrounds and experiences, potentially allowing for better diversity and representation in the human feedback provided. Future work could present comparisons or collaborations between these two methods.

### E.5 THE GRANULARITY OF SIMULATION

A key question that remains about audience simulation is whether there is a "sweet spot" for the level of detail a simulated audience should have. When we ask the LLM to generate stakeholder audiences, we do not specify the level of detail at which they should be generated. Consequently,

we notice that many of the comparisons (e.g., **Simulate** in Table 1) include a disclaimer about how the audience preference will depend on their personal details, suggesting that a more detailed description of generated audiences may improve performance. At the same time, a higher level of detail could also result in less accurate simulations due to the situation being less common. We note that even with the current setup, the LLM does not enter the first-person perspective for some simulations, and adding more details might make this behavior more frequent.

This effect is not just limited to the audiences, but also the details in the scenario itself. Various details of the real scenario can be included in the description fed into EGS. For instance, the plane crash example might be affected by whether there was a recent crash from the same company, or whether the weather conditions were bad that day, or even if people are generally more upset due to a global pandemic. We observe varying levels of hallucination in the generated candidates (see Appendix B.15 for a hallucination analysis), with many being partially caused by a lack of information provided in the scenario, where the model is forced to hallucinate in order to follow the generated advice. Thus, including more information does not only assist in simulation, but also reduces potential hallucinations in the model. Ultimately, there is a trade-off between trying to accurately replicate the scenario and making the described scenario easier for the LLM to reason about, which we leave to future work.

### E.6 LLMs AS A REPRESENTATION OF SHARED CULTURAL EXPERIENCE

By constructing a framework to help assist communication, we also make it possible to share information through the LLM's training data that may surpass individual, cultural, or geographical barriers. For example, a barista may not pay attention to a customer giving a coffee shop halfway across the globe a five star online review while describing their pleasant experience, but LLMs have the capability to take this information into account and synthesize a range of perspectives into the responses that they generate. Similarly, a person trying to solve a problem in their marriage might not have the habit of reading reddit forums on relationship advice, but LLMs might be able to take inspiration from these sources and use them to provide meaningful insights. In constructing this framework, we also hope to connect people with communication strategies that might not normally be available to them, not just helping them to improve their communication but also potentially helping them grow as communicators.

### E.7 BROADER IMPACT

Towards broader societal impact, EGS can help (1) with new or unfamiliar audiences where mental simulations are uncertain, and (2) reduce misunderstandings by taking the audience's point of view. EGS can also allow ideas to be shared in ways that are more acceptable, especially between groups that are divided in their opinions.

