# OpenReview forum: "Optimizing Interpersonal Communication by Simulating Audiences with Large Language Models"
_ICLR.cc/2024/Conference — Submitted to ICLR 2024_

### Official Review · Reviewer_hVxs · 2023-10-22

**Soundness:** 2 fair
**Presentation:** 3 good
**Contribution:** 3 good
**Rating:** 5
**Confidence:** 4

**Summary:**

This paper aims to improve inter-personal communication (general communication to achieve goals) by means of the proposed Explore-Generate-Simulate framework leveraging LLMs.
The framework 1) generates diverse pieces of advice that apply to a given scenario; 2) generates candidate responses based on each piece of advice; and 3) simulates how candidates will be received by possible audiences.
The authors compare their framework against other strategies like chain-of-thought across eight scenarios ranging from PR, marketing statements, and interpersonal domestic conflict.

**Strengths:**

1. The framework steps are intuitive and well grounded / explained. This can be seen as an interesting application of multi-agent conversations for LLMs grounded in theory about goal-oriented conversation. The paper could benefit from better defining e.g. advice and breaking down what constitutes good advice or any possible distinctions between how LLMs seem to generate responses conditioned on advice vs. how humans may take the advice. But overall the framework seems to make sense and is well explained.

2. The human study has good N (652) and evaluation of inter-rater reliability, which gives us confidence in the results. The experimental procedure could be better summarized in the main paper body - the detail in A.4 is in stark contrast to the brief explanation on page 6 of the paper itself.

3. The results in comparing EGS against zero-shot prompting and the version of Chain of Thought used here are encouraging, on both the human evaluation and simulated SHP evaluation.

**Weaknesses:**

1. My main concern here is that LLMs are highly sensitive to prompt engineering, wording, and order. This paper would benefit from a deeper discussion of how the prompts were developed (human-written, LLM-aided, etc.), any success or antipatterns noticed, and stability of results WRT prompts. As it stands the results are positive but

2. One specific concern is the quality of LLM reasoning over numeric values (e.g. in A.3 prompts weighting different stakeholders by asking the LLM to infer numeric ratings/weight for each stakeholder wrt the goal). When asking for multiple quantities in a prompt, it's unclear how effective each particular portion is, and the paper would benefit from an ablation or breakdown of generation quality in each stage.

3. The pilot study results (e.g. for number of pieces of advice) should be delineated in the main paper body to more strongly justify the design choices. Similarly, choices like "generat[ing] three candidates...to overcome any noise in generation" seems like empirical design and should at least be explained further. The framing of "You remember a piece of advice..." can also be better justified.

4. The demos are interesting but do bring up a concern about hallucinating information: for 4.2 for example, the friend names are hallucinated and things like the particular events (pool, break-up) also seem to be hallucinated. While this may seem like incidental information it's important to see an analysis of what hallucinations are typically generated under this framework for the LLM(s) chosen and whether they are material to the goal/conversation.

**Questions:**

See weaknesses - I would like to see some discussion of those points.

---

> ### Author Response · Authors · 2023-11-23
>
> Thank you Reviewer hVxs for your detailed review. We are elated that you find our framework steps intuitive and well-grounded, and we appreciate your attention towards the high quality of our human data and the promise that our results show.
>
> | The paper could benefit from better defining e.g. advice and breaking down what constitutes good advice or any possible distinctions between how LLMs seem to generate responses conditioned on advice vs. how humans may take the advice.
> Thank you for the suggestion. We have reworked the section where we describe the framework, including the section about exploring advice (3.1), to be more grounded in psychology literature that people recall useful advice or prior experiences when considering their next action. We have also better motivated the incorporation of advice into creating the message candidates (section 3.2), and we believe that the “inner voice” method from Park et al. (2023), combined with our framing of “remembering a piece of advice”, may be similar to how humans would typically interpret advice as well. However, there is still much work to be done investigating similarities and differences of SoTA LLMs and human behavior.
> | The experimental procedure could be better summarized in the main paper body - the detail in A.4 is in stark contrast to the brief explanation on page 6 of the paper itself.
> This is a good point - we have added a paragraph providing details on the rating interface that participants were given in Section 5.1. We also clarify the descriptions for the candidates and baselines that the humans are rating by providing examples in Appendix A.4 and the EGS prompt pipelines in Appendices A.3-A.7. Unfortunately, we have not had the luxury of space to move much more of the experiment description into the main text, but if you have any particular details you would like us to include, please let us know.
>
> | LLMs are highly sensitive to prompt engineering, wording, and order. This paper would benefit from a deeper discussion of how the prompts were developed, any success or antipatterns noticed, and stability of results WRT prompts.
>
> Thank you for this suggestion, we completely agree that the sensitivity to prompts is a concern with LLMs. First, our framework implementation randomly assigns candidates to be shown first or second in the pairwise comparisons. To complement this, we perform analysis on the order of the examples shown in the pairwise comparions (Appendix B.11) and demonstrate that whether a candidate shows up first or second does not bias whether it is favored.
> Second, when we constructed the prompts templates, we designed them to be very simple to minimizes possible biases towards or irregularities in performance across candidates. In all steps of the pipeline, we only prompt the LLM with the scenario description, the action it is to take, and the role it is supposed to simulate (if applicable). The only modification we make is to ask the model to generate reasoning before the final output following Wei et al. (2022), which is for performance as well as interpretability and sanity checks. We also add the full list of prompts in the order they are used, as well as examples, in Appendices A.3-A.7.
> Third, during development, we noticed only a few antipatterns w.r.t to the prompts: LLM outputs not obeying specific parsing rules (we cover this briefly in Appendix A.6), poorer agreement/performance on unorthodox advice (Appendices B.1 and B.3), and worse performance when trying to combine three (or presumably more) advice (Appendix B.9).
> Fourth, we notice a few successes in prompting as well: Prompting for unorthodox advice generates creative and unexpected outcomes that improve the ratings of downstream candidates (Appendix B.8); allowing for the conditioning on more than a single piece of advice improves the top-performing candidates but not as much in general (Appendix B.10); searching for conceptual advice instead of encouragement-type advice or irrelevant advice leads to significantly better performance (Appendix B.7).
> Fifth, we also highlight a list of general observations we make qualitatively, with specific examples included, in the new qualitative analysis (Section 4). The section contains six high level observations, two about Explore, two about Generate, and two about Simulate.
> Sixth and last, we conduct a preliminary analysis into the stability of results w.r.t. prompts in the SHP evaluation (Appendix C.2). Here, we replace each sentence in the redditor simulation prompt with a variant and find that performance is roughly equal or better than the original.

---

> > ### Author Response · Authors · 2023-11-23
> >
> > | One specific concern is the quality of LLM reasoning over numeric values - asking for multiple quantities in a prompt, it's unclear how effective each particular portion is, and the paper would benefit from an ablation or breakdown of generation quality in each stage.
> >
> > We cover the generation quality in each stage via the ablations and additional analyses mentioned in the previous section, and also provide examples of each step in the framework (including the prompts) in Appendices A.3-A.7.
> > One of the aspects that we have not yet covered is the numeric values of stakeholder weights that you mention. We would like to refer the reviewer to Appendix A.5.2, where the stakeholder descriptions, the weights assigned, and the justifications for the weights are provided for each scenario. Specifically, we would like to point out Plane Crash, Product Launch, and Dating App which are the scenarios with more than one stakeholder. Qualitatively, the weight balance seemed very reasonable in each. However, due to the subjective nature of weights and relative importance, it is hard to take the extra step to create a “gold-standard” weight distribution across stakeholders. This is particularly true when crowdworkers may not have the proper expertise and experience to provide well-informed weights, such as in the plane crash scenario.
> > It also might be worth noting that as LLMs improve, numerical reasoning on these tasks will become easier. As our framework is model-agnostic, its performance will also improve as the capabilities of the underlying LLMs get better.
> >
> >
> > | The pilot study results (e.g. for number of pieces of advice) should be delineated in the main paper body to more strongly justify the design choices. Similarly, choices like "generat[ing] three candidates...to overcome any noise in generation" seems like empirical design and should at least be explained further. The framing of "You remember a piece of advice..." can also be better justified.
> >
> > Thank you for the suggestions! We have included the numerical values for the pilot study results into Section 5.1 of the main paper to justify the design choices more clearly, and provide more details about the pilot in Appendix B.9.
> > For generating three candidates and framing “You remember a piece of advice”, we cite past works Liu and Shah (2023) and Park et al. (2023) respectively. These papers use similar techniques in their methods, which allow us to better justify our design decisions.
> >
> >
> > | it's important to see an analysis of what hallucinations are typically generated under this framework and whether they are material to the goal/conversation
> >
> > Thank you for your suggestion - we also believe that this analysis would be valuable, and we have included a full analysis in Appendix B.15, and point towards it in the discussion section (7.3). Perhaps as you expect, we observe that many of the hallucinations do not detract from the overall concepts that the candidates embody. Thus, when EGS is used in practice, these hallucinations would likely just be replaced by the user and serve as benign templates. For example, “Did Mike finally ace his pool game?'” can be replaced by anything interesting that the user remembers about their husband's friends.
> >
> > The only exceptions to this are when the advice themselves make assumptions that do not align with reality. For example, in the plane crash scenario, if the airline company does not have a long-standing reputation for safety, the corresponding candidates generated may be unusable. However, these hallucinations are rare, and incorporating more information into the scenario description or altering the advice using user controls (Appendix E.1) is a quick fix in the case that unusable advice is generated.

---

### Official Review · Reviewer_Dz3q · 2023-10-31

**Soundness:** 3 good
**Presentation:** 3 good
**Contribution:** 3 good
**Rating:** 6
**Confidence:** 3

**Summary:**

Human interpersonal communication can be difficult due to limited experience and time to make careful decisions. This paper studies the potential of using a language model to help humans communicate better. The paper proposes a EGS framework with exploration, generation, and simulation. The language model first explores by providing advice, including unorthodox advice. Next, the language model generates candidate messages that the human agent can use. Finally, the language model is used to simulate human behavior by evaluating the candidate messages. When simulating human behavior, an audience called stakeholders are generated, along with their importance weights. Finally, the stakeholders can evaluate the candidate messages and this can be used to derive an aggregate based on their answers and weights. The paper further proposes 8 scenarios that cover the ten fundamental processes of social interaction, such as social influence, social support, privacy management, and uncertainty management. By comparing with GPT-4 zero-shot and Chain of Thought, and by using human judgements as evaluation, the proposed EGS framework performed the best on 5 out of 8 scenarios. Finally, the paper discuss how the simulate step is well aligned with real-world web users, by using the Stanford Human Preferences dataset.

**Strengths:**

- Enhancing interpersonal communication may become one of the popular applications of language models, and it seems to be a potentially important research direction.
- The 8 scenarios cover diverse situations, and the paper discuss how they span the 10 fundamental processes of interpersonal communication.
- The proposed EGS framework is relatively simple and easy to understand.
- The paper introduces some unique ideas such as simulating stakeholders.
- The code and data is provided in the supplementary link.

**Weaknesses:**

- It would have been better to have some discussions about some of the recent advances in prompt engineering (post CoT), perhaps in Section 2. When I first read the manuscript, it wasn't clear to me why only the zero-shot and chain-of-thought baselines were used. My current understanding about the reason for not including other baselines is that other prompting methods such as Wang et al. (ICLR 2023) that also try to sample many candidates and then choose the best one does not work with the interpersonal communication tasks due to the lack of a fixed answer. Similar methods for open-ended generations (such as Jain et al. 2023) seem to also rely on the assumption that there is an underlying fixed answer and tries to look for semantically closer ones. The contributions of the paper will become more significant if we have this kind of discussion. I think it will also strengthen the motivation to study/focus on the interpersonal communication application, rather than focusing on general prompting methods.
- The stakeholder idea is interesting. It demonstrates how focusing on the interpersonal communication task is meaningful. However, it makes me wonder if it is meaningful to consider stakeholders for the scenarios after the first two. For the first two (plane crash, product launch), it seems to be a nice idea to have stakeholders. Figure 1 shows an example of generated stakeholders (sales, customer, media), but it would be better to have the generated stakeholders and their generated weights for all scenarios, perhaps in the appendix.
- Other variations of the EGS formulation: I wonder if we can improve the framework by simulating the stakeholders before the generate step. If we condition on each stakeholder and then generate candidates, will it generate better candidates compared to the case without conditioning on the stakeholders?
- Related to the "white lie" scenario, I feel the "negative use" paragraph in Section 7 can be discussed in more detail. For example, the language model can generate messages that may include more serious lies that superficially improve the communication but with more societal/ethical harm.

Wang et al. (ICLR 2023): Self-consistency improves chain of thought reasoning in language models

Jain et al. (2023): Self-consistency for open-ended generations

**Questions:**

Minor comments and questions:
- Park et al. (2023a) and Park et al. (2023b) seem to be the same paper.
- Ref error in Section A.4: (Appendix ??)
- It would be helpful to have a table with 10 fundamental processes on the columns and 8 scenarios on the rows, and where each cells indicate if the scenario includes the fundamental process or not with a check mark.
- Is it more precise if we say "zero-shot Chain of Thought" instead of "Chain of Thought"? The prompt example shown in Table 5 seems like it is the zero-shot version of CoT.

**Comment after rebuttal period:**

Thank you for updating the paper and answering my questions. I currently do not have further questions.

---

> ### Author Response · Authors · 2023-11-23
>
> Thank you reviewer Dz3q for your review, we appreciate you for going through the paper in such detail. We are excited that you share our vision that studying the application of LLMs to interpersonal communication is an important research direction.
>
>
> | discussions about some of the recent advances in prompt engineering, such as self-consistency
>
> Thank you for your suggestion, we completely agree! In addition to self-consistency being reliant on the assumption of an underlying fixed answer, we performed a new ablation on unorthodox advice that suggests that the self-consistency method’s motivating intuition that better answers = higher probability does not hold in the interpersonal communication case. We have added a discussion of prompt engineering past work the extended related work (Appendix D), and we point to it in Section 2. We add a justification of the baseline selection in Appendix A.7.1, and point to it in Section 5.1. Details on the new ablation for unorthodox advice are in Appendix B.8.
>
>
> | I think it will also strengthen the motivation to study/focus on the interpersonal communication application, rather than focusing on general prompting methods.
>
> We have edited various parts of the paper to reflect this suggestion:
> In the prompt engineering paragraph of the extended past work (Appendix D), we specifically differentiate our work from other prompting methods by highlighting that it is focused on the interpersonal communication application.
> We add a paragraph on episodic future thinking from psychology to the past work section (Section 2), drawing connections between our work and existing literature on the human process of simulating future outcomes. We also connect our framework back to this concept in the discussion (Section 7.4).
> We add various psychology and computer science literature to ground the claims in our introduction and methods sections (Sections 1 and 3).
> Please let us know if you feel we can do more in any other areas.
>
>
> | The stakeholder idea is interesting. It demonstrates how focusing on the interpersonal communication task is meaningful. However, it makes me wonder if it is meaningful to consider stakeholders for the scenarios after the first two.
>
> We completely agree - multiple stakeholders/audiences is a feature of EGS that can have variable impact depending on the problem context. As our target is to assist users in all communication scenarios, we believe multiple stakeholders is an important piece of the framework. In cases where only one audience is needed, the LLM does not generate extraneous audiences, and nothing is lost.
>
>
> | it would be better to have the generated stakeholders and their generated weights for all scenarios, perhaps in the appendix.
>
> We have added a comprehensive list of the stakeholders of each scenario, their weights, and their full descriptions in Appendix A.5. Please feel free to take a look!
>
>
> | I wonder if we can improve the framework by simulating the stakeholders before the generate step. If we condition on each stakeholder and then generate candidates, will it generate better candidates compared to the case without conditioning on the stakeholders?
>
> This is an interesting suggestion! We conducted a quick experiment to see how candidates generated based on both stakeholders and advice, compared to those generated with just advice. Specifically, for the plane crash scenario, we compare the best candidate against six new candidates. These contain three candidates conditioned on the best candidate’s advice set and the “general public” stakeholder, and three candidates conditioned on the same advice pair but on the “victims’ families” stakeholder. Across three simulated comparisons per pair and audience profile for 36 comparisons total, we find that none of the new candidates are favored compared to the original best candidate, suggesting that conditioning on specific audiences does not perform substantially better than conditioning only on advice. We add a detailed description of this experiment, including prompts + design choices, the new generated candidates, and the pairwise comparison results in Appendix B.12.
> It is worth noting that these comparisons were done by GPT-4, and even though we have shown that agreement between humans and GPT-4 in the scenario were high, it is still possible that humans would rate these new candidates differently. Additionally, your suggestion also inspires the more general question of: Which steps of EGS or interpersonal communication in general benefit most from considering audience reactions? We think is a very interesting future direction to study.

---

> > ### Author Response · Authors · 2023-11-23
> >
> > | I feel the "negative use" paragraph in Section 7 can be discussed in more detail. For example, the language model can generate messages that may include more serious lies that superficially improve the communication but with more societal/ethical harm.
> >
> > We completely agree, and we have added to the discussion in Section 7.3 to include a discussion on this. It is worth noting that in our new qualitative analysis, we observe that GPT-4 tends to avoid lying in the ‘white lie during date’ scenario, indicating that safety tuning is beneficial towards avoiding negative outcomes. However, it remains that models can be jailbroken and models without safety measures may be used with EGS, and so awareness must be made surrounding these issues as well.
> >
> > | helpful to have a table with 10 fundamental processes on the columns and 8 scenarios on the rows, and where each cells indicate if the scenario includes the fundamental process or not with a check mark.
> >
> > We made this exact table! (Table 5, Appendix A.2) Thank you for this suggestion.
> >
> >
> > | Is it more precise if we say "zero-shot Chain of Thought" instead of "Chain of Thought"?
> >
> > Yes - we have changed the paper to reflect this. We have also added reasoning in Appendix A.7.1 (pointed to in Section 5.1) to justify why we avoid few-shot CoT:
> > Difficulty of identifying exemplar utterances in interpersonal communication
> > Limited relevance any particular exemplar would have to the entire problem domain.
> >
> > All the other minor comments and questions are fixed. Thank you for your careful review!

---

### Official Review · Reviewer_GN7N · 2023-11-05

**Soundness:** 3 good
**Presentation:** 3 good
**Contribution:** 2 fair
**Rating:** 6
**Confidence:** 3

**Summary:**

This paper introduces a new framework for communicative message generation under different scenarios. It adopts a three-step pipeline: explore (explore key advice for each scenario) → generate (generate initial messages based on different advice combinations) → simulate (simulate audiences with different perspectives to select the final best message). The authors evaluate this framework in eight scenarios. They collect human selections of candidates and ratings over messages generated by EGS and other baselines. The experimental results show that EGS agrees with human selection and gets ratings higher than other baselines. This framework is further applied to a real Internet user simulation, which validates its generality to real-world scenarios.

**Strengths:**

1. Leveraging multiple LLMs to serve different roles to complete a given task is a promising research direction.
2. The authors validate this framework through a comprehensive design of experiments, which encompasses comparisons with human selection, various baseline models, and practical applications in real-world scenarios.
3. The EGS framework prompts LLMs to generate advice and audiences corresponding to the current task, which enables it to be easily generalized to different scenarios.

**Weaknesses:**

1. Scenario design: it is not clear to me how these eight scenarios are chosen based on the 10 fundamental processes of interpersonal communication, and why it is representative of all possible social tasks.
2. The initial generation results bottleneck the final generation. As the example shown in Section 4.3, both candidates may demonstrate some advantages. It would be interesting if it could aggregate the valid points in both candidates and then decide the final generation.

**Questions:**

1. In step 2, when saying “iterates over subsets of advice”, how these subsets are formed?
2. How the prompts are designed for the CoT baseline?
3. Experimental results demonstration:

    a. For Table 2, the authors choose “>0.6” as a high agreement. However, it would be better if a baseline is included to clarify how a “high agreement” is decided.

    b. Similarly in Table 3, it would be better to mark the significance between comparisons.

    c. In Table 5, what do the results represent? I think it is the accuracy of selecting the correct upvote, but it is not stated in the paper.

---

> ### Author Response · Authors · 2023-11-23
>
> We thank Reviewer GN7N for their clear and thoughtful review. We are encouraged that the reviewer found that using LLMs to serve different roles to be a promising research direction, and our experiments to be comprehensive.
>
> | how the eight scenarios are chosen based on the 10 fundamental processes of interpersonal communication, and why it is representative of all possible social tasks
>
> Generalizability and representativeness was something that we carefully considered throughout our project, especially since the domain of general interpersonal communication is impressively large. We would like to share a list of steps that we have taken to try to make our evaluation (including the scenarios) as cohesive as possible, which is available in the first point of the general comment.
> Though we do not believe nor claim that our experiments are fully representative of all social tasks, we believe that we have taken significant steps in trying to ensure the generality of our analysis, while the former is much more difficult to achieve.
>
>
> | The initial generation results bottleneck the final generation. It would be interesting if it could aggregate the valid points in both candidates and then decide the final generation.
>
> Your suggestion to use the pairwise comparisons to create new candidates is very interesting! We conducted some preliminary investigations that we report below, and we have added a more detailed description to Appendix B.14, and add a shoutout in the main paper in Section 3.3.
>
> Method:
> After the simulate step, we take the top k candidates and their pairwise comparisons, and ask GPT-4 to generate a new candidate that performs better based on the comparisons. For more details including the specific prompts, please see Appendix B.14. We also mention this briefly in the main paper as an alternative when introducing the aggregation of the simulated comparisons.
>
> As this method would require inserting all relevant pairwise comparisons into the new candidate generation prompt, it is better to use a smaller k to be able to check manually that the comparisons are actually taken into consideration. In our preliminary investigation, we use k=2 as a simple example.
>
> Results:
> We find that the new candidates try to incorporate information across both candidates, and are thus much longer than their original counterparts. For example, in the white lie during date scenario, the two original candidates are:
>
> "Wow, I must say that style isn't easy to pull off, but you certainly can! You just light up in whatever you wear and that joy is contagious. It's not everyone who can wear a jacket like that and make it work. You make it feel like more than just a jacket, you bring it to life!"
>
> "Wow, I love how that jacket brings out the sparkle in your eyes, and I can see how happy it makes you feel. It certainly adds a unique flair to your whole look."
>
> And the new candidate:
>
> "Wow, that jacket sure is unique! And trust me, it's not everyone who can wear something like that and bring it to life. The way it brings out the sparkle in your eyes is really captivating. Above everything, what I love the most is seeing how happy it makes you feel. Your joy really is contagious, and it lights up your whole look!"
>
> Examples are in Appendix B.14.
>
> Speculatively, this often makes their utterances too lengthy for the scenario. Future investigations can be done to see if these outcomes are indeed less preferred by human raters, while also exploring how LLMs can better combine message candidates in a concise manner.
>
> More generally, there are many options after the simulate step to combine scores into a final candidate message. EGS can be viewed as a search + filter framework, and the best-performing advice and/or candidates can be interpreted and aggregated by the user as they see fit. We also add a brief evaluation on various alternative aggregation methods motivated by cognitive science literature (Gates et al., 2020), including maximin, maxsum, and inequality aversion (see Appendix B.13).
>
>
> | In step 2, when saying “iterates over subsets of advice”, how these subsets are formed?
>
> Initially, subsets of advice were formed by taking the powerset of the set of advice. To restrict the advice space to a reasonable and efficient size, we limited advice sets to the power set of cardinality 2. This decision is backed up by a prelimiary study suggesting that > 2 pieces of advice was less effective, which we have included briefly in Section 5.1 and in more detail in Appendix B.9.

---

> ### Author Response · Authors · 2023-11-23
>
> | How the prompts are designed for the CoT baseline?
>
> The prompt is designed to be very simple to minimizes possible biases towards any specific candidates. Specifically, we only prompt the LLM with the scenario description, the action, and the communicator’s role. We also ask the model to generate reasoning before the final output, following Wei et al. (2022).
>
> The prompts themselves are structured as follows:
> “[Scenario description]
> What would you [insert action] if you were [insert communicator]? Please reason about the situation before providing your answer. Provide your answer in this format:
> Reasoning: [text]
> Answer: [text]”
>
> An example for the product launch scenario:
> “{Scenario description}
> What would you say if you were the CEO? Please reason about the situation before providing your answer. Provide your answer in this format:
> Reasoning: [text]
> Answer: [text]”
>
> We have also added a full list of prompts in Appendix A.7.2.
>
> | Experimental results demonstration:
>
> a. For Table 2, the authors choose “>0.6” as a high agreement. However, it would be better if a baseline is included to clarify how a “high agreement” is decided.
>
> We agree with the reviewer that “high” agreement is not well defined, especially when we defined our own agreement metric. We replace all instances of “high agreement” with “significant agreement”, and back up this claim with two new agreement analyses comparing the mean scores of winners and losers of pairwise comparisons in the Simulate step.
>
> One is a multi-level model where the scenario is treated as a random effect, and we find a significant fixed effect for the winner vs. loser of the simulated audience pairwise judgements (see Section 5.3). The other finds that in individual scenarios, the winners also obtain significantly higher scores in a paired samples t-test across five scenarios and a subset of a sixth scenario (see Table 3, which was previously Table 2). Both of these analyses use a metric complimentary to the previous metric in the original, which we justify in the paragraph before the third metric (our original percentage agreement) is introduced in Section 5.3.
>
> b. Similarly in Table 3, it would be better to mark the significance between comparisons.
>
> We have added statistical significance testing comparing EGS with both baselines using bootstrapping with 10,000 samples to the result. Averaging over the eight scenarios, we find that EGS significantly outperforms both GPT-4 zero shot and CoT baselines (see Table 2, which was previously Table 3).
>
> c. In Table 5, what do the results represent? I think it is the accuracy of selecting the correct upvote, but it is not stated in the paper.
>
> Yes, this is correct! We have added the relevant details in the first sentence of the second paragraph of Section 6.

---

### Official Review · Reviewer_V2zC · 2023-11-10

**Soundness:** 3 good
**Presentation:** 3 good
**Contribution:** 3 good
**Rating:** 5
**Confidence:** 4

**Summary:**

The paper proposes the idea of incorporating LLM to communicate better in various scenarios when the user has a specific goal in mind and wants to communicate by employing a strategy (advice) such that the predefined goal would be achieved. Authors show that LLMs are used in generating different advices or strategies, subsequently different outputs and they are capable of simulating the outcome of each generation from the perspective of various audiences.

**Strengths:**

The idea of incorporating LLMs to communicate better and use them not only for generation but also for simulating the outcome of each generation on different audiences seems promising, as the nature of communications sometimes can be very complicated and indeed finding the best strategy would be much more challenging. With the good performance of LLMs in many domains authors claim that they can be reliable for easing the communications. This paper tries to address such kind of problem by merely focusing on the ability of LLMs.

**Weaknesses:**

Even though the idea seems to be promising my main concern is regarding the shortage of evidence in proving and showing the framework's performance. The approach is tested on a limited set of scenarios which does not provide a strong proof of the model's performance in different domains (and its generalizability). One possible benchmark could be the negotiation conversations to check what percentage of the time the proposed approach will be able to win the negotiation. However in section 6 the stimulate step is assessed on SHP dataset, it would be nice to have more fine-grained study on the type of the domain/user preferences and personalities and their connection with the framework's performance.

**Questions:**

How much the proposed framework for improving the interpersonal communication is affected by the underlying LLM's social norms. Since the proposed method relies on off-the-shelf LLMs, it is important to investigate the outcome of their generations in different cultures or on people with different personalities. In other word, relying directly on LLMs that have been trained on data with specific social norms in the background should have different outcomes which urges a comprehensive study in various cultures/domains.

In table 2, how do we assure that the outcome of EGS is significantly better than the other baselines? have you done any significance testing?

---

> ### Author Response · Authors · 2023-11-23
> **Response to Reviewer V2zC**
>
> We thank Reviewer V2zC for acknowledging the difficulty of the domain of problems we propose a solution to. We agree with you that generalizability is a key challenge in our work, and we would be glad to share a list of steps we have taken throughout the project to address this, available in the first point of the general comment.
>
> | One possible benchmark could be the negotiation conversations to check what percentage of the time the proposed approach will be able to win the negotiation
>
> The bargaining metric you propose – evaluating the percentage of the time the approach wins the negotiation – is quite similar to what we measure when we ask the crowdworkers to evaluate our generated candidates & baselines. Specifically, we ask, “How likely would you be willing to negotiate the price of the vase with Jill?”, and the scale (0) Definitely not willing ... (5) Neutral ... (10) Definitely willing (see Section 5.1, second-to-last paragraph). Here, the metric is softer than whether the candidate would win the negotiation, but it serves a similar purpose. The other 7 scenarios also each have a unique metric just like this one, which can be found in the bulleted list in Appendix A.8.
>
>
> | on SHP dataset, it would be nice to have more fine-grained study on the type of the domain/user preferences and personalities and their connection with the framework's performance
>
> While it is difficult to exactly identify the preferences and personalities of individual Redditors, we can use each subreddit rules and guidelines to better understand the preferences of Redditors who browse specific domains. As discussed at the end of Section 6, we found that some domains such as asksocialscience dictate that comments “must be supported by citations”. This preference is reflected in the performance of the framework with the Funny setting being significantly worse than that of the Default setting. For the askculinary domain, the subreddit rule states that posts and comments should be “specific” and “detailed”, but there are no rules to force all comments to be strictly informative. Consequently, the Funny setting performed better on this domain.
>
> To further understand how the domain/user preferences are connected with the framework’s performance, we have also added a more fine-grained study on the SHP dataset. Specifically, we perturb the Funny Redditor prompt to reflect different user preferences and personalities, which results in a total of 6 different Redditor profiles. We conducted this experiment on the askculinary domain, and the new prompts can be found in Table 18.
>
> We found that two out of the four new profiles performed identically to the original Funny prompt while the other two saw significant improvements. The highest performing one asks the LLM to prefer “niche” topics, so the performance increase could be attributed to the “specific” and “detailed” rules of the askculinary subreddit. More analysis can be found in Appendix C. We thus found that the performance of the framework depends on the preferences of the domain and the alignment between the used prompt and user preferences.
>
>
> | it is important to investigate the outcome of their generations in different cultures or on people with different personalities.
>
> We share your concern about our framework being affected by underlying social norms captured by the LLM.  We agree that thorough investigation is necessary with respect to people of different cultures and personalities. As EGS is model-agnostic, this becomes a large undertaking outside the scope of this paper, since such a study would need to identify and define new metrics and evaluation protocols, given the EGS framework. We plan to follow-up with a more comprehensive effort in future work. In the meantime, we have added this shared concern in the discussion (see Section 7.3, last paragraph).
>
> It is worth noting that one benefit of EGS is its interpretability and ease of selecting alternatives (see Section 7.1). Thus, it makes it easy to analyze through EGS for any potential biases that the underlying LLM may contain. Furthermore, it also makes it easy to find alternatives in case the user is not satisfied with the recommendation.
>
>
> | In table 2, how do we assure that the outcome of EGS is significantly better than the other baselines? have you done any significance testing?
>
> Yes! We have added statistical significance testing to the result, where we compare EGS with both baselines using bootstrapping with 10000 samples each. Averaging over the eight scenarios, we find that EGS significantly outperforms both GPT-4 zero shot and CoT baselines (see Table 2).

---

### Author Response · Authors · 2023-11-23
**General Comment to all Reviewers**

We thank the reviewers for your careful reviews of our paper. All your feedback is very much appreciated and has made the paper better. Here, we summarize a few pieces of feedback that multiple reviewers touched on.

1. Generalizability to arbitrary interpersonal communication/social tasks
We agree with you that generalizability is a key challenge in our work, and it has been a focus of ours throughout the project. To this end, we would like to share a list of steps that we have taken to ensure generalizability:

a. To provide broad coverage of the space of human communications, we constructed our scenarios based on the ten fundamental processes of interpersonal communication (Berger and Roloff, 2019), a framework grounded in psychology literature. We provide a mapping from scenarios to these ten fundamental processes in Table 7 of Appendix 2.

b. When constructing our scenarios, we also made sure to cover different types and modalities of interpersonal communication. We cover broadcast, written, and spoken communication to diversify the settings and level of uncertainties the communicator (and EGS) have of the audiences to be simulated. We also distribute our scenarios between professional (first 4) and personal (last 4) relationships.

c. We collect a novel dataset of 12,180 human preferences from Prolific crowdworkers in order to assist our evaluation on EGS. In order to conduct reliable investigations on our framework, we sacrifice amount of scenarios for the quantity of evaluations per scenario, collecting 20+ judgements each for tens of candidates and baselines while keeping costs reasonable. This allowed us to test and claim that our method statistically significantly outperforms both baselines, and that it agrees with human judgements. We believe this was a necessary trade-off compared to expanding the number of scenarios to tackle generalizability to arbitrary interpersonal communication or social tasks.

d. We also conduct analyses on the SHP dataset, which consists of real user behavior on Reddit, to test our framework on a wider range of communication. We believe this supports our claim towards the generalizability of the performance of EGS.

e. We conducted a new suite of three ablation studies to show that 1) searching for conceptual advice outperforms both encouragement-type and unrelated advice (Appendix B.7), 2) unorthodox advice improves downstream message candidates (Appendix B.8), 3) allowing multiple advice to be conditioned on at once improves scores of top candidates (Appendix B.10), and much more. This provides additional evidence that the framework contains design decisions well-justified by empirical results.

2. We would like to bring the reviewers’ attention to a few portions of the paper that we have edited significantly since the original submission, in order of importance:

a. Agreement between humans and GPT-4 (Section 5.3): We add two new metrics, including a multi-level model where the scenario is treated as a random effect and a paired samples t-test within each scenario. Both analyses find significance in the agreement between human ratings and GPT-4 pairwise comparisons, which is something we did not have before.

b. Qualitative Analysis (Section 4): We create a new table containing a snapshot of each step of Explore-Generate-Simulate for the same scenario, and perform qualitative analyses. We also describe some new general observations about each of these steps that we were only able to obtain by eye.

c. Why is default LLM prompting insufficient? (right before Section 3.1): This paragraph gives a better motivation for why we investigate our research questions, and highlights some key challenges that existing methods face.

d. Past work - Simulations in the human mind: This relates our work to the cognitive science concept of “episodic future thinking”, or simulating the future using our minds. We cover existing research in psychology and neuroscience and conceptually relate these to LLMs in terms of simulating episodic future thought.

e. Discussion + extended discussion: We believe many of the topics discussed are interesting, and a majority are completely new. These include extensions to multi-turn conversations, interpretability, user controls, applications to counterfactual reasoning, whether there is an optimal simulation granularity, and more.

f. We have also added many new ablation studies and additional analyses into the appendix, and which are used to support various points in the main paper.

---

### Meta-Review · Area_Chair_qToo · 2023-12-11

**Metareview:**

This paper proposes using GPT-4 to improve communication. Given a particular communication goal, this approach follows a 3 step process that prompts GPT-4 to generate the content in three stages. First, it generates a few communication strategies. Then, candidate responses are based on those strategies.  And finally, rating them based on simulated audience reaction. User studies show that the responses generated with this strategy are rated more highly than zero-shot prompt or chain-of-thought prompts to GPT-4.

The discussion period has been productive. The changes made have improved clarity, and additional experiments have strengthened the findings. This is an application area where language models can be very helpful, as acknowledged by reviewers.

However, this paper would be a better fit for another venue that is dedicated to communication or human-computer interaction. The discussion of baselines and methodology is not precise from a machine learning perspective. The paper does not discuss other challenges that the proposed approach introduces, for example, increased inference time. The application, though important and interesting, and the sequential prompting strategy seem narrow for this venue. The paper would be a better fit for a different venue focused on communication or human-computer interaction so that its value can properly be recognized.

**Justification For Why Not Higher Score:**

The application, though important and interesting, and the sequential prompting strategy seem narrow for this venue. The paper might be a better fit for a different venue focused on communication or human-computer interaction so that its value can properly be recognized.

**Justification For Why Not Lower Score:**

N/A

---

### Decision · Program_Chairs · 2024-01-16

Reject